# Modulate Your Spectrum in Self-Supervised Learning

**Xi Weng**[*1], **Yunhao Ni**[*1], **Tengwei Song**[1], **Jie Luo**[1],
**Rao Muhammad Anwer**[2], **Salman Khan**[2], **Fahad Shahbaz Khan**[2], **Lei Huang** [✉ 1]
[1]SKLCCSE, Institute of Artificial Intelligence, Beihang University, Beijing, China
[2]Mohamed bin Zayed University of Artificial Intelligence, UAE

## Abstract

Whitening loss offers a theoretical guarantee against feature collapse in self-supervised learning (SSL) with joint embedding architectures. Typically, it involves a hard whitening approach, transforming the embedding and applying loss to the whitened output. In this work, we introduce Spectral Transformation (ST), a framework to modulate the spectrum of embedding and to seek for functions beyond whitening that can avoid dimensional collapse. We show that whitening is a special instance of ST by definition, and our empirical investigations unveil other ST instances capable of preventing collapse. Additionally, we propose a novel ST instance named IterNorm with trace loss (INTL). Theoretical analysis confirms INTL's efficacy in preventing collapse and modulating the spectrum of embedding toward equal-eigenvalues during optimization. Our experiments on ImageNet classification and COCO object detection demonstrate INTL's potential in learning superior representations. The code is available at https://github.com/winci-ai/INTL.

## 1 Introduction

Self-supervised learning (SSL) via joint embedding architectures to learn visual representations has made significant progress over the last several years (Bachman et al., 2019; He et al., 2020; Chen et al., 2020a; Chen & He, 2021; Bardes et al., 2022; Oquab et al., 2023), almost outperforming their supervised counterpart on many downstream tasks (Liu et al., 2021; Jaiswal et al., 2020; Ranasinghe et al., 2022). This paradigm addresses to train a dual pair of networks to produce similar embeddings for different views of the same image (Chen & He, 2021). One main challenge with the joint embedding architectures is how to prevent a *collapse* of the representation, in which the two branches ignore the inputs and produce identical and constant outputs (Chen & He, 2021). A variety of methods have been proposed to successfully avoid *collapse*, including contrastive learning methods (Wu et al., 2018; He et al., 2020; Saunshi et al., 2019) that attract different views from the same image (positive pairs) while pull apart different images (negative pairs), and non-contrastive methods (Grill et al., 2020; Chen & He, 2021) that directly match the positive targets without introducing negative pairs.

The collapse problem is further generalized into *dimensional collapse* (Hua et al., 2021; Jing et al., 2022) (or *informational collapse* (Bardes et al., 2022)), where the embedding vectors only span a lower-dimensional subspace and would be highly correlated. In this case, the covariance matrix of embedding has certain zero eigenvalues, which degenerates the representation in SSL. To prevent *dimensional collapse*, a theoretically motivated paradigm, called whitening loss, is proposed by minimizing the distance between embeddings of positive pairs under the condition that embeddings from different views are whitened (Ermolov et al., 2021; Hua et al., 2021). One typical implementation of whitening loss is hard whitening (Ermolov et al., 2021; Weng et al., 2022) that designs whitening transformation over mini-batch data and imposes the loss on the whitened output (Ermolov et al., 2021; Hua et al., 2021; Weng et al., 2022). We note that the whitening transformation is a function over embedding during forward pass, and modulates the spectrum of embedding implicitly during backward pass when minimizing the objective. This raises questions whether there exist other functions over embedding can avoid collapse? If yes, how the function affects the spectrum of embedding?

This paper proposes spectral transformation (ST), a framework to modulate the spectrum of embedding in joint embedding architecture. ST maps the spectrum of embedding to a desired distribution during forward pass, and modulates the spectrum of embedding by implicit gradient update during backward pass (Figure 1).

---

[*]equal contribution  [✉]corresponding author (huangleiAI@buaa.edu.cn).

This framework provides a way to seek for functions beyond whitening transformation that can avoid dimensional collapse. We show that whitening transformation is a special instance of ST using a power function by definition, and there exist other power functions that can avoid dimensional collapse by our empirical investigation (see Section 3.2 for details). We demonstrate that IterNorm (Huang et al., 2019), an approximating whitening method by using Newton's iterations (Bini et al., 2005; Ye et al., 2020), is also an instance of ST, and show that IterNorm

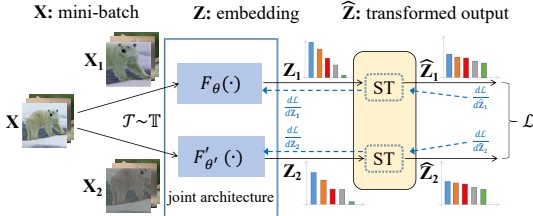

Figure 1: The framework using spectral transformation (ST) to modulate the spectrum of embedding in joint embedding architecture for SSL.

with different iteration number corresponds to different ST (see Section 3.2.2 for details). We further theoretically characterize how the spectrum evolves as the increasing of iteration number of IterNorm.

We empirically observe that IterNorm suffers from severe dimensional collapse and mostly fails to train the model in SSL unexpectedly, unlike its benefits in approximating whitening for supervised learning (Huang et al., 2019). We thus propose IterNorm with trace loss (INTL), a simple solution to address the failure of IterNorm, by adding an extra penalty on the transformed output. Moreover, we theoretically demonstrate that INTL can avoid dimensional collapse, and reveal its mechanism in modulating the spectrum of embedding to be equal-eigenvalues. We conduct comprehensive experiments and show that INTL is a promising SSL method in practice. Our main contributions are summarized as follows:

- We propose spectral transformation, a framework to modulate the spectrum of embedding and to seek for functions beyond whitening that can avoid dimensional collapse. We show there exist other functions that can avoid dimensional collapse by empirical observation and intuitive explanation.

- We propose a new instance of ST, called IterNorm with trace loss (INTL). We theoretically prove that INTL can avoid collapse and modulate the spectrum of embedding towards an equal-eigenvalue distribution during the course of optimization.

- INTL's experimental performance on standard benchmarks showcases its high promise as a practical SSL method, consistently achieving or surpassing state-of-the-art methods, even when utilizing a relatively small batch size.

## 2 RELATED WORK

Our work is related to the SSL methods that address the feature collapse problem when using joint embedding architectures.

**Contrastive learning** prevents collapse by attracting positive samples closer, and spreading negative samples apart (Wu et al., 2018; Ye et al., 2019). In these methods, negative samples play an important role and need to be well designed (Oord et al., 2018; Bachman et al., 2019; Henaff, 2020). MoCos (He et al., 2020; Chen et al., 2020b) build a memory bank with a momentum encoder to provide consistent negative samples, while SimCLR (Chen et al., 2020a) addresses that more negative samples in a batch with strong data augmentations perform better. Our proposed INTL can avoid collapse and work well without negative samples. Additionally, recent work (Zhang et al., 2023) explores the use of data augmentation for contrastive learning through spectrum analysis to enhance performance, while our paper focuses on developing a novel non-contrastive method to prevent collapse under standard augmentation.

**Non-contrastive learning** can be categorized into two groups: *asymmetric methods* and *whitening loss*. *Asymmetric methods* employ asymmetric network architectures to prevent feature collapse without the need for explicit negative pairs (Caron et al., 2018; 2020; Li et al., 2021; Grill et al., 2020; Chen & He, 2021). For instance, BYOL (Grill et al., 2020) enhances network stability by appending a predictor after the online network and introducing momentum into the target network. SimSiam (Chen & He, 2021) extends BYOL and emphasizes the importance of stop-gradient to prevent trivial solutions. Other advancements in this realm include cluster assignment prediction using the Sinkhorn-Knopp algorithm (Caron et al., 2020) and the development of asymmetric pipelines with self-distillation losses for Vision Transformers (Caron et al., 2021). However, it remains unclear how these asymmetric networks effectively prevent collapse without the inclusion of negative pairs. This has sparked debates surrounding topics such as batch normalization (BN)(Fetterman & Albrecht, 2020; Tian et al., 2020b; Richemond et al., 2020) and stop-gradient(Chen & He, 2021; Zhang et al., 2022a). Despite preliminary efforts to analyze training dynamics (Tian et al., 2021) and establish

connections between non-contrastive and contrastive methods (Tao et al., 2022; Garrido et al., 2023), the exact mechanisms behind these methods remain an ongoing area of research. In our work, we address the more intricate challenge of dimensional collapse and theoretically demonstrate that our INTL method effectively prevents this issue, offering valuable insights into mitigating feature collapse in various scenarios.

*Whitening loss* is a theoretically motivated paradigm to prevent dimensional collapse (Ermolov et al., 2021). One typical implementation of whitening loss is *hard whitening* that designs whitening transformation over mini-batch data and imposes the loss on the whitened output. The designed whitening transformation includes batch whitening in W-MSE (Ermolov et al., 2021) and Shuffled-DBN (Hua et al., 2021), channel whitening in CW-RGP (Weng et al., 2022), and the combination of both in Zero-CL (Zhang et al., 2022b). Our proposed ST generalizes whitening transformation and provides a frame to modulate the spectrum of embedding. Our INTL can improve these work in training stability and performance, by replacing whitening transformation with IterNorm (Huang et al., 2019) and imposing an additional trace loss on the transformed output. Furthermore, we theoretically show that our proposed INTL modulates the spectrum of embedding to be equal-eigenvalues.

Another way to implement whitening loss is *soft whitening* that imposes a whitening penalty as regularization on the embedding, including Barlow Twins (Zbontar et al., 2021), VICReg (Bardes et al., 2022) and CCA-SSG (Zhang et al., 2021). Different from these works, our proposed INTL imposes the trace loss on the approximated whitened output, providing equal-eigenvalues modulation on the embedding.

There are also theoretical works analyzing how dimensional collapse occurs (Hua et al., 2021; Jing et al., 2022) and how it can be avoided by using whitening loss (Hua et al., 2021; Weng et al., 2022). The recent works (He & Ozay, 2022; Ghosh et al., 2022) further discuss how to characterize the magnitude of dimensional collapse, and connect the spectrum of a representation to a power law. They show the coefficient of the power law is a strong indicator for the effects of the representation. Different from these works, our theoretical analysis presents a new thought in demonstrating how to avoid dimensional collapse, which provides theoretical basis for our proposed INTL.

## 3 SPECTRAL TRANSFORMATION BEYOND WHITENING

### 3.1 PRELIMINARY AND NOTATION

**Joint embedding architectures.** Let $\mathbf{x}$ denote the input sampled uniformly from a set of images $\mathbb{D}$, and $\mathbb{T}$ denote the set of data transformations available for augmentation. We consider a pair of neural networks $F_\theta$ and $F'_{\theta'}$, parameterized by $\theta$ and $\theta'$ respectively. They take as input two randomly augmented views, $\mathbf{x}^{(1)} = \mathcal{T}_1(\mathbf{x})$ and $\mathbf{x}^{(2)} = \mathcal{T}_2(\mathbf{x})$, where $\mathcal{T}_{1,2} \in \mathbb{T}$; and they output the *embedding* $\mathbf{z}^{(1)} = F_\theta(\mathbf{x}^{(1)})$ and $\mathbf{z}^{(2)} = F'_{\theta'}(\mathbf{x}^{(2)})$. The networks are trained with an objective function that minimizes the distance between embeddings obtained from different views of the same image:

$$\mathcal{L}(\mathbf{x}, \theta) = \mathbb{E}_{\mathbf{x} \sim \mathbb{D}, \, \mathcal{T}_{1,2} \sim \mathbb{T}} \; \ell\big(F_\theta(\mathcal{T}_1(\mathbf{x})), F'_{\theta'}(\mathcal{T}_2(\mathbf{x}))\big). \tag{1}$$

where $\ell(\cdot, \cdot)$ is a loss function. The mean square error (MSE) of $L_2-$normalized vectors as $\ell(\mathbf{z}^{(1)}, \mathbf{z}^{(2)}) = \|\frac{\mathbf{z}^{(1)}}{\|\mathbf{z}^{(1)}\|_2} - \frac{\mathbf{z}^{(2)}}{\|\mathbf{z}^{(2)}\|_2}\|_2^2$ is usually used as the loss function (Chen & He, 2021). This loss is also equivalent to the negative cosine similarity, up to a scale of $\frac{1}{2}$ and an optimization irrelevant constant (Chen & He, 2021). This architecture is also called *Siamese Network* (Chen & He, 2021), if $F_\theta = F'_{\theta'}$. Another variant distinguishes the networks into target network $F'_{\theta'}$ and online network $F_\theta$, and updates the weight $\theta'$ of target network through exponential moving average (EMA) (Chen et al., 2020b; Grill et al., 2020) over $\theta$ of online network.

**Feature collapse.** While minimizing Eqn. 1, a trivial solution known as *complete collapse* could occur such that $F_\theta(\mathbf{x}) \equiv \mathbf{c}, \; \forall \mathbf{x} \in \mathbb{D}$. Moreover, a weaker collapse condition called *dimensional collapse* can be easily arrived, for which the projected features collapse into a low-dimensional manifold. To express dimensional collapse more mathematically, we refer to dimensional collapse as the phenomenon that one or certain eigenvalues of the covariance matrix of feature vectors degenerate to 0. Therefore, we can determine the occurrence of dimensional collapse by observing the spectrum of the covariance matrix.

**Whitening loss.** To address the collapse problem, whitening loss (Ermolov et al., 2021) is proposed to minimize Eqn. 1, under the condition that *embeddings* from different views are whitened. Whitening loss provides theoretical guarantee in avoiding (dimensional) collapse, since the embedding is whitened with all axes decorrelated (Ermolov et al., 2021; Hua et al., 2021). Ermolov *et*

*al.* (Ermolov et al., 2021) propose to whiten the mini-batch embedding $\mathbf{Z} \in \mathbb{R}^{d \times m}$ using batch whitening (BW) (Huang et al., 2018; Siarohin et al., 2019) and impose the loss on the whitened output $\widehat{\mathbf{Z}} \in \mathbb{R}^{d \times m}$, given the mini-batch inputs $\mathbf{X}$ with size of $m$, as follows:

$$\min_{\theta} \mathcal{L}(\mathbf{X}; \theta) = \mathbb{E}_{\mathbf{X} \sim \mathbb{D}, \, \mathcal{T}_{1,2} \sim \mathbb{T}} \|\widehat{\mathbf{Z}}^{(1)} - \widehat{\mathbf{Z}}^{(2)}\|_F^2$$

$$with \; \widehat{\mathbf{Z}}^{(v)} = \Sigma^{-\frac{1}{2}} \mathbf{Z}^{(v)}, \, v \in \{1, 2\}, \tag{2}$$

where $\Sigma = \frac{1}{m} \mathbf{Z}\mathbf{Z}^T$ is the covariance matrix of embedding[1]. $\Sigma^{-\frac{1}{2}}$ is called the whitening matrix, and is calculated either by Cholesky decomposition in (Ermolov et al., 2021) or by eigen-decomposition in (Hua et al., 2021). *E.g.*, zero-phase component analysis (ZCA) whitening (Huang et al., 2018) calculates $\Sigma^{-\frac{1}{2}} = \mathbf{U}\Lambda^{-\frac{1}{2}}\mathbf{U}^T$, where $\Lambda = \mathrm{diag}(\lambda_1, \ldots, \lambda_d)$ and $\mathbf{U} = [\mathbf{u}_1, ..., \mathbf{u}_d]$ are the eigenvalues and associated eigenvectors of $\Sigma$, *i.e.*, $\mathbf{U}\Lambda\mathbf{U}^T = \Sigma$. One intriguing result shown in (Weng et al., 2022) is that hard whitening can avoid collapse by only constraining the embedding $\mathbf{Z}$ to be full-rank, but not whitened.

We note that the whitening transformation is a function over embedding $\mathbf{Z}$ during forward pass, and modulates the spectrum of embedding $\mathbf{Z}$ implicitly during backward pass when minimizing MSE loss imposed on the whitened output. This raises a question of whether there are other functions over embedding $\mathbf{Z}$ that can avoid collapse? If yes, how the function affects the spectrum of embedding $\mathbf{Z}$?

## 3.2 SPECTRAL TRANSFORMATION

In this section, we extend the whitening transformation to spectral transformation (ST), a more general view to characterize the modulation on the spectrum of embedding, and empirically investigate the interaction between the spectrum of the covariance matrix of $\widehat{\mathbf{Z}}$ and collapse of the SSL model.

**Definition 1.** *(**Spectral Transformation**) Given any unary function $g(\cdot)$ in the definition domain $\lambda(\mathbf{Z}) = \{\lambda_1, \lambda_2, \ldots, \lambda_d\}$. Drawing an analogy with whitening, $g(\cdot)$ on the covariance matrix $\Sigma$ of embedding $\mathbf{Z}$ is defined as $g(\Sigma) = \mathbf{U}g(\Lambda)\mathbf{U}^T$, where $g(\Lambda) = diag(g(\lambda(\mathbf{Z})))$. We denote the transformation matrix of ST as $\Phi_{ST} = g(\Sigma)$, so that the output of ST is calculated by $\widehat{\mathbf{Z}} = \Phi_{ST}\mathbf{Z} = \mathbf{U}g(\Lambda)\mathbf{U}^T\mathbf{Z}$ and the covariance matrix of $\widehat{\mathbf{Z}}$ is $\Sigma_{\widehat{\mathbf{Z}}} = \frac{1}{m}\widehat{\mathbf{Z}}\widehat{\mathbf{Z}}^T = \mathbf{U}\Lambda g^2(\Lambda)\mathbf{U}^T$.*

Based on Definition 1, ST is an abstract framework until $g(\cdot)$ is determined, and its essence is mapping the spectrum $\lambda(\mathbf{Z})$ to $\lambda(\widehat{\mathbf{Z}}) = \left\{\lambda_1 g^2(\lambda_1), \lambda_2 g^2(\lambda_2), \ldots, \lambda_d g^2(\lambda_d)\right\}$. When applied in the context of self-supervised learning, the loss function for ST remains the same as Eqn 2, with the only difference being that $\widehat{\mathbf{Z}}$ is determined by $g(\cdot)$. Meanwhile, the optimization direction for the embedding spectrum can also be determined when employing gradient-based methods. That is, what spectrum of embedding will be modulated to be during the course of training.

**Can we unveil the potential of ST?** Our ST framework exhibits uncertainty and diversity, allowing $g(\cdot)$ to adopt the guise of any single-variable function within the defined domain, including power functions, exponential functions, iterative functions, and more. Whitening, on the other hand, is a special and successful instance within ST, where $g(\cdot)$ takes the form of a power function $g(\lambda) = \lambda^{-\frac{1}{2}}$. This naturally prompts two questions: 1. Could there be other functions, akin to whitening, capable of preventing collapse within the ST framework? 2. If yes, how the function works and affects the spectrum of embedding $\mathbf{Z}$?

### 3.2.1 SPECTRAL TRANSFORMATION USING POWER FUNCTIONS

With these questions in mind, we embark on a deeper exploration of the mechanics extending beyond whitening, considering a more comprehensive transformation $g(\lambda) = \lambda^{-p}, p \in (-\infty, +\infty)$ for ST. Based on Definition. 1, this comprehensive power transformation is mapping the spectrum $\lambda(\mathbf{Z})$ to $\lambda(\widehat{\mathbf{Z}}) = \left\{\lambda_1^{1-2p}, \lambda_2^{1-2p}, \ldots, \lambda_d^{1-2p}\right\}$.

**Empirical observation.** Initially, we conduct experiments on a 2D dataset, varying the parameter $p$, and visualize the outputs of the toy models as depicted in Figure 2(a). Our observations indicate that the toy model tends to perform well in avoiding collapse when $p$ falls within the neighborhood of 0.5, specifically in the range of 0.45 to 0.55. However, as $p$ gradually deviates from 0.5, collapse becomes more pronounced. Subsequently, we extend our experiments to real-world datasets to validate these findings. The results presented in Figure 2(b) align with the previously observed phenomena. When $p$ is set to either 0.45 or 0.55, the model maintains high evaluation performance, similar to that

---

[1]The embedding is usually centralized by performing $\mathbf{Z} := \mathbf{Z}(\mathbf{I} - \frac{1}{m}\mathbf{1} \cdot \mathbf{1}^T)$ for whitening, and we assume $\mathbf{Z}$ is centralized in this paper for simplifying discussion.

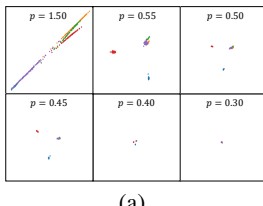 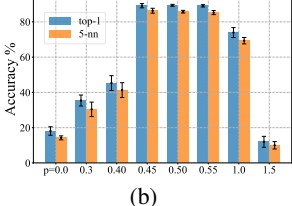 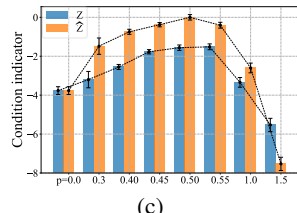

|  |  |  |
| --- | --- | --- |
| (a) | (b) | (c) |

Figure 2: Investigate ST using power functions. We choose several $p$ from 0 to 1.5. We show (a) the visualization of the toy model output; (b) top-1 and 5-nearest neighbors (5-nn) accuracy on CIFAR-10; (c) condition indicator of embedding $\mathbf{Z}$ and transformed output $\widehat{\mathbf{Z}}$ on CIFAR-10. We use the inverse of the condition number (IoC) in logarithmic scale with base 10 ( $lgIoC = lgc^{-1} = lg\frac{\lambda_d}{\lambda_1}$ ) as the condition indicator. The results on CIFAR-10 are obtained through training with ResNet-18 for 200 epochs and averaged over five runs, with standard deviation shown as error bars. We show the details of experimental setup in *Appendix* D. Similar phenomena can be observed when using other datasets (*e.g.*, ImageNet) and other networks (*e.g.*, ResNet-50).

of whitening ($p = 0.5$). This discovery suggests that within the ST framework, there exist other functions capable of successfully preventing collapse, which answers the first question.

For the second question, as illustrated in Figure 2(c), it becomes evident that when $p$ lies in the vicinity of $0.5$, the embedding showcases a more well-conditioned spectrum, characterized by a smaller condition number (larger IoC). However, when $p$ significantly deviates from $0.5$, the spectrum of embedding loses its well-conditioned attributes, closely aligning with the occurrence of embedding collapse. This statement asserts that if $g(\cdot)$ is effective in preventing collapse within the ST framework, it will result in the modulation of the embedding spectrum towards a well-conditioned state.

**Intuitive explanation.** We note that (Weng et al., 2022) implied that whitening loss in Eqn 2 can be decomposed into two asymmetric losses $\mathcal{L} = \frac{1}{m}\|\phi(\mathbf{Z}^{(1)})\mathbf{Z}^{(1)} - (\widehat{\mathbf{Z}}^{(2)})_{st}\|_F^2 + \frac{1}{m}\|\phi(\mathbf{Z}^{(2)})\mathbf{Z}^{(2)} - (\widehat{\mathbf{Z}}^{(1)})_{st}\|_F^2$, where $\phi(\mathbf{Z})$ refers to the whitening matrix of $\mathbf{Z}$, $st$ represents the stop gradient operation, and $\widehat{\mathbf{Z}}$ denotes the whitening output. Each asymmetric loss can be viewed as an online network to match a whitened target $\widehat{\mathbf{Z}}$. As a more generalized form of whitening, our ST can also extend this decomposition to the loss function. As depicted in Figure 2(c), when $p$ falls within the range of $0.45$ to $0.55$, $\widehat{\mathbf{Z}}$ exhibits a well-conditioned spectrum, with each eigenvalue approaching 1. In such instances, $\widehat{\mathbf{Z}}$ serves as an ideal target for $\phi(\mathbf{Z})\mathbf{Z}$ to match, enabling the embedding $\mathbf{Z}$ to learn a favorable spectrum to prevent collapse. Conversely, when $p$ deviates significantly from $0.5$, the spectrum of the transformed output loses its well-conditioned characteristics, with $\widehat{\mathbf{Z}}$ becoming a detrimental target, ultimately leading to the collapse of the embedding.

### 3.2.2 IMPLICIT SPECTRAL TRANSFORMATION USING NEWTON'S ITERATION

However, utilizing the power function $g(\lambda) = \lambda^{-p}$ (where $p$ is approximately 0.5) within our ST framework is not without its drawbacks. One issue is the potential for numerical instability when computing eigenvalues $\lambda$ and eigenvectors $\mathbf{U}$ via eigen-decomposition, particularly when the covariance matrix is ill-conditioned (Paszke et al., 2019). We provide comprehensive experiments and analysis in *Appendix* D.3 to validate the presence of this problem in SSL.

Naturally, if we could implement a spectral transformation that can modulate the spectrum without the need for explicit calculation of $\lambda$ or $\mathbf{U}$, this issue could be mitigated. In fact, we take note of an approximate whitening method called iterative normalization (IterNorm) (Huang et al., 2019), which uses Newton's iteration to address the numerical challenges associated with batch whitening in supervised learning. Specifically, given the centered embedding $\mathbf{Z}$, the iteration count $T$, and the trace-normalized covariance matrix $\Sigma_N = \Sigma/tr(\Sigma)$, IterNorm performs Newton's iteration as follows.

$$\begin{cases} \mathbf{P}_0 = \mathbf{I} \\ \mathbf{P}_k = \frac{1}{2}(3\mathbf{P}_{k-1} - \mathbf{P}_{k-1}^3 \Sigma_N), & k = 1, 2, ..., T. \end{cases} \tag{3}$$

The whitening matrix $\Sigma^{-\frac{1}{2}}$ is approximated by $\Phi_T = \mathbf{P}_T/\sqrt{tr(\Sigma)}$ and we have the whitened output $\widehat{\mathbf{Z}} = \Phi_T\mathbf{Z}$. When $T \to +\infty$, $\Phi_T \to \Sigma^{-\frac{1}{2}}$ and the covariance matrix of $\widehat{\mathbf{Z}}$ will be an identity matrix. Here, we theoretically show that IterNorm is also an instance of spectral transformation as follows.

**Theorem 1.** *Define one-variable iterative function $f_T(x)$, satisfying*

$$f_{k+1}(x) = \frac{3}{2}f_k(x) - \frac{1}{2}xf_k^{\,3}(x), k \geq 0; \ f_0(x) = 1.$$

*The mapping function of IterNorm is $g(\lambda) = f_T(\frac{\lambda}{tr(\Sigma)})/\sqrt{tr(\Sigma)}$. Without calculating $\lambda$ or $\mathbf{U}$, IterNorm implicitly maps $\forall \lambda_i \in \lambda(\mathbf{Z})$ to $\widehat{\lambda}_i = \frac{\lambda_i}{tr(\Sigma)} f_T^2(\frac{\lambda_i}{tr(\Sigma)})$.*

The proof is provided in *Appendix* B.1. For simplicity, we define the T-whitening function of IterNorm $h_T(x) = x f_T^2(x)$, which obtains the spectrum of transformed output. Based on the fact that the covariance matrix of transformed output will be identity when $T$ of IterNorm increases to infinity (Bini et al., 2005), we thus have

$$\forall \lambda_i > 0, \lim_{T \to \infty} h_T(\frac{\lambda_i}{tr(\Sigma)}) = 1. \tag{4}$$

Different iteration numbers $T$ of IterNorm imply different T-whitening functions $h_T(\cdot)$. It is interesting to analyze the characteristics of $h_T(\cdot)$.

**Proposition 1.** *Given $x \in (0,1)$, $\forall T \in \mathbb{N}$ we have $h_T(x) \in (0,1)$ and $h_T'(x) > 0$.*

The proof is shown in *Appendix* A.1. Proposition 1 states $h_T(x)$ is a monotone increasing function for $x \in (0,1)$ and its range is also in $(0,1)$. Since $\frac{\lambda_i}{tr(\Sigma)} \in (0,1)$, $\forall \lambda_i > 0$, we have

$$\forall T \in \mathbb{N}, \lambda_i > \lambda_j > 0 \Longrightarrow 1 > \widehat{\lambda}_i > \widehat{\lambda}_j > 0. \tag{5}$$

Formula 5 indicates that IterNorm maps all non-zero eigenvalues to $(0,1)$ and preserves monotonicity.

**Proposition 2.** *Given $x \in (0,1)$, $\forall T \in \mathbb{N}$, we have $h_{T+1}(x) > h_T(x)$.*

The proof is shown in *Appendix* A.2. Proposition 2 indicates that IterNorm gradually stretches the eigenvalues towards one as the iteration number $T$ increases. This property of IterNorm theoretically shows that the spectrum of $\widehat{\mathbf{Z}}$ will have better condition if we use a larger iteration number $T$ of IterNorm.

In summary, our analyses theoretically show that IterNorm gradually stretches the eigenvalues towards one as the iteration number $T$ increases, and the smaller the eigenvalue is, the larger $T$ is required to approach one.

## 4 ITERATIVE NORMALIZATION WITH TRACE LOSS

It is expected that IterNorm, as a kind of spectral transformation, can avoid collapse and obtain good performance in SSL, due to its benefits in approximating whitening for supervised learning (Huang et al., 2019). However, we empirically observe that IterNorm suffers severe dimensional collapse and mostly fails to train the model in SSL (we postpone the details in Section 4.2.). Based on the analyses in Section 3.2 and 3.2.2, we propose a simple solution by adding an extra penalty named *trace loss* on the transformed output $\widehat{\mathbf{Z}}$ by IterNorm to ensure a well-conditioned spectrum. It is clear that the sum of eigenvalues of $\Sigma_{\widehat{\mathbf{Z}}}$ is less than or equal to $d$, we thus propose a trace loss that encourages the trace of $\Sigma_{\widehat{\mathbf{Z}}}$ to be its maximum $d$, when $d \leq m$. In particular, we design a new method called *IterNorm with trace loss* (INTL) for optimizing the SSL model as[2]:

$$\min_{\theta \in \Theta} \quad INTL(\mathbf{Z}) = \sum_{j=1}^{d} (1 - (\Sigma_{\widehat{\mathbf{Z}}})_{jj})^2, \tag{6}$$

where $\mathbf{Z} = F_\theta(\cdot)$ and $\widehat{\mathbf{Z}} = IterNorm(\mathbf{Z})$. Eqn. 6 can be viewed as an optimization problem over $\theta$ to encourage the trace of $\widehat{\mathbf{Z}}$ to be $d$.

### 4.1 THEORETICAL ANALYSIS

In this section, we theoretically prove that INTL can avoid collapse, and INTL modulates the spectrum of embedding towards an equal-eigenvalue distribution during the course of optimization.

Note that $\Sigma_{\widehat{\mathbf{Z}}}$ can be expressed using the T-whitening function $h_T(\cdot)$ as $\Sigma_{\widehat{\mathbf{Z}}} = \sum_{i=1}^{d} h_T(x_i)\mathbf{u}_i \mathbf{u}_i^T$, where $x_i = \lambda_i / tr(\Sigma) \geq 0$ and $\sum_{i=1}^{d} x_i = 1$. When the range of $F_\theta(\cdot)$ is wide enough, the optimization

---

[2]The complete loss function of INTL is $\mathcal{L}_{INTL} = \mathcal{L}_{MSE} + \beta \cdot \mathcal{L}_{trace}$, where the coefficient $\beta$ is fixed across all datasets and architectures, and its determination is elaborated in 'Algorithm of INTL' of *Appendix* C. To simplify the discussion, we omit the $\mathcal{L}_{MSE}$ term here, without compromising the validity.

Figure 3: Investigate the effectiveness of IterNorm with and without trace loss. We train the models on CIFAR-10 with ResNet-18 for 100 epochs. We apply IterNorm with various iteration numbers $T$, and show the results with (solid lines) and without (dashed lines) trace loss respectively. (a) The spectrum of the embedding $\mathbf{Z}$; (b) The spectrum of the transformed output $\widehat{\mathbf{Z}}$; (c) The top-1 accuracy. (d) indicates that IterNorm (without trace loss) suffers from numeric divergence when using a large iteration number, e.g. $T = 9$. It is noteworthy that when $T \geq 11$, the loss values are all **NAN**, making the model unable to be trained. Similar phenomena can be observed when using other datasets (*e.g.*, ImageNet) and other networks (*e.g.*, ResNet-50).

problem over $\theta$ (Eqn. 6) can be transformed as the following optimization problem over $\mathbf{x}$ (Eqn. 7) without changing the optimal value (please see *Appendix* B.2 for the details of derivation):

$$
\begin{cases}
\min_{\mathbf{x}} & INTL(\mathbf{x}) = \sum_{j=1}^{d} \left( \sum_{i=1}^{d} [1 - h_T(x_i)] u_{ji}^2 \right)^2 \\
s.t. & \sum_{i=1}^{d} x_i = 1 \\
& x_i \geq 0, i = 1, \cdots, d,
\end{cases}
\tag{7}
$$

where $u_{ji}$ is the $j$-th elements of vector $\mathbf{u}_i$. In this formulation, we can prove that our proposed INTL can theoretically avoid collapse, as long as the iteration number $T$ of IterNorm is larger than zero.

**Theorem 2.** *Let* $\mathbf{x} \in [0,1]^d$, $\forall T \in \mathbb{N}_+$, $INTL(\mathbf{x})$ *shown in Eqn. 7 is a strictly convex function.* $\mathbf{x}^* = [\frac{1}{d}, \cdots, \frac{1}{d}]^T$ *is the unique minimum point as well as the optimal solution to* $INTL(\mathbf{x})$.

The proof is provided in *Appendix* B.2. Based on Theorem 2, INTL modulates the spectrum of embedding to be equal-eigenvalues during the backward pass, which provides a theoretical guarantee to avoid dimensional collapse.

**Connection to hard whitening.** Hard whitening methods, like W-MSE (Ermolov et al., 2021) and shuffle-DBN (Hua et al., 2021), design a whitening transformation over each view and minimize the distances between the whitened outputs from different views. This mechanism modulates the covariance matrix of *embedding* to be full-rank (Weng et al., 2022). Our INTL designs an approximated whitening transformation using IterNorm and imposes an additional trace loss penalty on the (approximately) whitened output, which modulates the covariance matrix of *embedding* having equal eigenvalues.

**Connection to soft whitening.** Soft whitening methods, like Barlow-Twins (Zbontar et al., 2021) and VICReg (Bardes et al., 2022) directly impose a whitening penalty as a regularization on the *embedding*. This modulates the covariance matrix of the *embedding* to be identity (with a **fixed** scalar $\gamma$, *e.g.*, $\gamma \mathbf{I}$). Our INTL imposes the penalty on the transformed output, but can be viewed as implicitly modulating the covariance matrix of the *embedding* to be identity with a **free** scalar (*i.e.*, having equal eigenvalues).

Intuitively, INTL modulates the spectrum of embedding to be equal-eigenvalues during the backward pass, which is a stronger constraint than hard whitening (the full-rank modulation), but a weaker constraint than soft whitening (the whitening modulation). This preliminary but new comparison provides a new way to understand the whitening loss in SSL.

### 4.2 EMPIRICAL ANALYSIS

In this section, we empirically show that IterNorm-only and trace-loss-only fail to avoid collapse, but IterNorm with trace loss can well avoid collapse.

**IterNorm fails to avoid collapse.** In theory, IterNorm can map all non-zero eigenvalues to approach one, with a large enough $T$. In practice, it usually uses a fixed $T$, and it is very likely to encounter small eigenvalues during training. In this case, IterNorm cannot ensure the transformed output has a well-conditioned spectrum (Figure 3(b)), which potentially results in dimensional collapse. One may use a large $T$, however, IterNorm will encounter numeric divergence upon further increasing the iteration number $T$, even though it has converged. *E.g.*, IterNorm suffers from numeric divergence in Figure 3(d) when using $T = 9$, since the maximum eigenvalue of whitened output is around

Table 1: Classification top-1 accuracy of a linear classifier and a 5-nearest neighbors classifier for different loss functions and datasets. The table is mostly inherited from solo-learn (da Costa et al., 2022). All methods are based on ResNet-18 with two augmented views generated from per sample and are trained for 1000-epoch on CIFAR-10/100 with a batch size of 256 and 400-epoch on ImageNet-100 with a batch size of 128.

| Method | CIFAR-10 | | CIFAR-100 | | ImageNet-100 | |
|---|---|---|---|---|---|---|
| | top-1 | 5-nn | top-1 | 5-nn | top-1 | 5-nn |
| SimCLR (Chen et al., 2020a) | 90.74 | 85.13 | 65.78 | 53.19 | 77.64 | 65.78 |
| MoCo V2 (Chen et al., 2020b) | **92.94** | 88.95 | 69.89 | 58.09 | 79.28 | 70.46 |
| BYOL (Grill et al., 2020) | 92.58 | 87.40 | 70.46 | 56.46 | 80.32 | 68.94 |
| SwAV (Caron et al., 2020) | 89.17 | 84.18 | 64.88 | 53.32 | 74.28 | 63.84 |
| SimSiam (Chen & He, 2021) | 90.51 | 86.82 | 66.04 | 55.79 | 78.72 | 67.92 |
| W-MSE (Ermolov et al., 2021) | 88.67 | 84.95 | 61.33 | 49.65 | 69.06 | 58.44 |
| Shuffled-DBN (Hua et al., 2021) | 91.17 | 88.95 | 66.81 | 57.27 | 75.27 | 67.21 |
| DINO (Caron et al., 2021) | 89.52 | 86.13 | 66.76 | 56.24 | 74.92 | 64.30 |
| Barlow Twins (Zbontar et al., 2021) | 92.10 | 88.09 | **70.90** | 59.40 | 80.16 | 72.14 |
| VICReg (Bardes et al., 2022) | 92.07 | 87.38 | 68.54 | 56.32 | 79.40 | 71.94 |
| Zero-CL (Zhang et al., 2022b) | 90.81 | 87.51 | 70.33 | 59.21 | 79.26 | 71.18 |
| CW-RGP (Weng et al., 2022) | 92.03 | 89.67 | 67.78 | 58.24 | 76.96 | 68.46 |
| **INTL (ours)** | 92.60 | **90.03** | 70.88 | **61.90** | **81.68** | **73.46** |

Table 2: Comparisons on ImageNet linear classification with various training epochs. All methods are based on ResNet-50 backbone with two augmented views generated from per sample. EMA represents Exponential Moving Average. Given that one of the objectives of SSL methods is to achieve high performance with small batch sizes, it's worth noting that our INTL performs effectively when trained with small batch sizes, such as 256 and 512.

| Method | Batch size | EMA | 100 eps | 200 eps | 400 eps | 800 eps |
|---|---|---|---|---|---|---|
| SimCLR | 4096 | No | 66.5 | 68.3 | 69.8 | 70.4 |
| SwAV | 4096 | No | 66.5 | 69.1 | 70.7 | 71.8 |
| | 512 | No | 65.8 | 67.9 | - | - |
| SimSiam | 256 | No | 68.1 | 70.0 | 70.8 | 71.3 |
| W-MSE | 512 | No | 65.1 | 66.4 | - | - |
| Shuffled-DBN | 512 | No | 65.2 | - | - | - |
| Barlow Twins | 2048 | No | 67.7 | - | 72.5 | **73.2** |
| Zero-CL | 1024 | No | 68.9 | - | **72.6** | - |
| CW-RGP | 512 | No | 67.1 | 69.6 | - | - |
| **INTL (ours)** | 512 | No | **69.5** | **71.1** | 72.4 | 73.1 |
| MoCo v2 | 256 | Yes | 67.4 | 69.9 | 71.0 | 72.2 |
| BYOL | 4096 | Yes | 66.5 | 70.6 | 73.2 | **74.3** |
| **INTL (ours)** | 256 | Yes | **69.2** | **71.5** | **73.7** | **74.3** |

$10^7$, significantly large than 1 (we attribute to the numeric divergence, since this result goes against Proposition 1 and 2, and we further validate it by monitoring the transformed output). It is noteworthy that when $T \geq 11$, the loss values are all **NAN**, making the model unable to be trained. These problems make IterNorm difficult to avoid dimensional collapse in practice.

**The Synergy between IterNorm and trace loss.** IterNorm in combination with trace loss demonstrates significant differences compared to IterNorm-only. Our experimental results, as shown in Figure 3(a), empirically confirm that INTL effectively prevents dimensional collapse, aligning with the findings of Theorem 2. INTL encourages the uniformity of eigenvalues within the covariance matrix of the embedding **Z**, resulting in well-conditioned spectra for the transformed output (Figure 3(b)) and impressive evaluation performance (Figure 3(c)), even when the iteration count $T$ is as low as 1. To further evaluate the performance of trace-loss-only, we conducte experiments under the same setup. Without IterNorm, trace-loss-only achieves a top-1 accuracy of only $16.15\%$, indicating significant collapse. Therefore, the efficacy of INTL, as well as the attainment of an optimal solution characterized by equal eigenvalues, is a result of the synergy between IterNorm and trace loss.

## 5 EXPERIMENTS ON STANDARD SSL BENCHMARK

In this section, we conduct experiments on standard SSL benchmarks to validate the effectiveness of our proposed INTL. We first evaluate the performance of INTL for classification on CIFAR-10/100 (Krizhevsky, 2009), ImageNet-100 (Tian et al., 2020a), and ImageNet (Deng et al., 2009). Then we evaluate the effectiveness in transfer learning, for a pre-trained model using INTL. We provide the full PyTorch-style algorithm in *Appendix* C as well as details of implementation and computational overhead in *Appendix* E.

Table 3: Transfer Learning. All competitive unsupervised methods are based on 200-epoch pre-training on ImageNet (IN). The table are mostly inherited from (Chen & He, 2021). Our INTL is performed with 3 random seeds, with mean and standard deviation reported.

| Method | COCO detection | | | COCO instance seg. | | |
|---|---|---|---|---|---|---|
| | $AP_{50}$ | AP | $AP_{75}$ | $AP_{50}$ | AP | $AP_{75}$ |
| Scratch | 44.0 | 26.4 | 27.8 | 46.9 | 29.3 | 30.8 |
| Supervised | 58.2 | 38.2 | 41.2 | 54.7 | 33.3 | 35.2 |
| SimCLR | 57.7 | 37.9 | 40.9 | 54.6 | 33.3 | 35.3 |
| MoCo v2 | 58.8 | 39.2 | 42.5 | 55.5 | 34.3 | 36.6 |
| BYOL | 57.8 | 37.9 | 40.9 | 54.3 | 33.2 | 35.0 |
| SwAV | 57.6 | 37.6 | 40.3 | 54.2 | 33.1 | 35.1 |
| SimSiam | 57.5 | 37.9 | 40.9 | 54.2 | 33.2 | 35.2 |
| W-MSE (repro.) | 60.1 | 39.2 | 42.8 | 56.8 | 34.8 | 36.7 |
| Barlow Twins | 59.0 | 39.2 | 42.5 | 56.0 | 34.3 | 36.5 |
| **INTL (ours)** | $\mathbf{60.9}_{\pm 0.08}$ | $\mathbf{40.7}_{\pm 0.09}$ | $\mathbf{43.7}_{\pm 0.17}$ | $\mathbf{57.3}_{\pm 0.08}$ | $\mathbf{35.4}_{\pm 0.05}$ | $\mathbf{37.6}_{\pm 0.14}$ |

## 5.1 EVALUATION FOR CLASSIFICATION

**Evaluation on small and medium size datasets.**    We initially train and perform linear evaluation of INTL using ResNet-18 as the backbone on CIFAR-10/100 (Krizhevsky, 2009) and ImageNet-100 (Tian et al., 2020a). We strictly adhere to the experimental settings outlined in solo-learn (da Costa et al., 2022) for these datasets. As depicted in Table 1, INTL achieves remarkable results, with a top-1 accuracy of $92.60\%$ on CIFAR-10, $70.88\%$ on CIFAR-100, and $81.68\%$ on ImageNet-100. These results are on par with or even surpass the state-of-the-art methods as reproduced by solo-learn. Furthermore, when employing a 5-nearest neighbors classifier, INTL outperforms other baselines by a significant margin, underscoring its capacity to learn superior representations.

**Evaluation on ImageNet.**    To further assess the versatility of INTL, we train it using a ResNet-50 backbone and evaluate its performance using the standard linear evaluation protocol on ImageNet. The results, presented in Table 2, demonstrate the effectiveness of INTL, achieving top-1 accuracy of $69.5\%$, $71.1\%$, $72.4\%$, and $73.1\%$ after pre-training for 100, 200, 400, and 800 epochs, respectively. We observe that our INTL performs even better when utilized in conjunction with the Exponential Moving Average (EMA) technique, as employed in BYOL and MoCo. This combination yielded a top-1 accuracy of $74.3\%$ after 800 epochs of training.

## 5.2 TRANSFER TO DOWNSTREAM TASKS

We examine the representation quality by transferring our pre-trained model to other tasks, including COCO (Lin et al., 2014) object detection and instance segmentation. We use the baseline of the detection codebase from MoCo (He et al., 2020) for INTL. The results of baselines shown in Table 3 are mostly inherited from (Chen & He, 2021). We observe that INTL performs much better than other state-of-the-art approaches on COCO object detection and instance segmentation, which shows the great potential of INTL in transferring to downstream tasks.

## 5.3 ABLATION STUDY

We conducte a comprehensive set of ablation experiments to assess the robustness and versatility of our INTL in *Appendix* F. These experiments cover various aspects, including batch sizes, embedding dimensions, the use of multi-crop augmentation, semi-supervised training, the choice of Vision Transformer (ViT) backbones and adding trace loss to other methods. Through these experiments, we gain valuable insights into how INTL performs under different conditions and configurations, shedding light on its adaptability and effectiveness in diverse scenarios. The results collectively reinforce the notion that INTL is a robust and flexible self-supervised learning method capable of delivering strong performance across a wide range of settings and data representations. Notably, our INTL achieved a remarkable top-1 accuracy of $76.6\%$ on ImageNet linear evaluation with ResNet-50 when employing multi-crop augmentation, surpassing even the common supervised baseline of $76.5\%$.

## 6 CONCLUSION

In this paper, we proposed spectral transformation (ST) framework to modulate the spectrum of embedding and to seek for functions beyond whitening that can avoid dimensional collapse. Our proposed IterNorm with trace loss (INTL) is well-motivated, theoretically demonstrated, and empirically validated in avoiding dimension collapse. Comprehensive experiments have shown the merits of INTL for achieving state-of-the-art performance for SSL in practice. We showed that INTL modulates the spectrum of embedding to be equal-eigenvalues during the backward pass, which is a stronger constraint than hard whitening (the full-rank modulation), but a weaker constraint than soft whitening (the whitening modulation). This preliminary but new results provides a potential way to understand and compare SSL methods.

## ACKNOWLEDGMENTS

This work was partially supported by the National Science and Technology Major Project under Grant 2022ZD0116310, National Natural Science Foundation of China (Grant No. 62106012), the Fundamental Research Funds for the Central Universities.

## REPRODUCIBILITY STATEMENT

To ensure the reproducibility and comprehensiveness of our paper, we have included an appendix comprising six main sections. These sections serve various purposes:

- Appendix A contains detailed proofs for the propositions presented in our work.
- Appendix B provides in-depth proofs for the theorems introduced in our research.
- Appendix C offers a comprehensive view of the INTL algorithm, including detailed formulas and PyTorch-style code for implementation.
- Appendix D elaborates on the settings used in our analytical experiments, with reference to Figure 2 and Figure 3.
- Appendix E furnishes insights into the implementation details and computational intricacies of experiments conducted on standard SSL benchmarks, as discussed in Section 5.
- Finally, Appendix F encompasses a comprehensive set of ablation experiments, assessing the robustness and versatility of our INTL method across various scenarios.

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

## A  PROOFS OF PROPOSITION

### A.1  PROOF OF PROPOSITION.1

**Proposition 1.**  Given $x \in (0, 1)$, $\forall T \in \mathbb{N}$ we have $h_T(x) \in (0, 1)$ and $h'_T(x) > 0$.

*Proof.* We know the iterative function $f_T(x)$ satisfies

$$f_{k+1}(x) = \frac{3}{2}f_k(x) - \frac{1}{2}x{f_k}^3(x), k \geq 0; f_0(x) = 1 \tag{8}$$

We define $h_T(x) = x{f_T}^2(x)$. When $x = 1$, it is easy to verify $\forall T \in \mathbb{N}$, $h_T(1) = f_T(1) = 1$. We first prove $f_T(x) > 0$ and $h'_T(x) > 0$ by mathematical induction.

(1) When $T = 0$, we have $f_0(x) = 1 > 0$, and $h_0(x) = x$, $h'_0(x) = 1 > 0$.

(2) Assuming it holds when $T = k$, we have $f_k(x) > 0$ and $h'_k(x) > 0$. Based on $h'_k(x) = f_k(x)[f_k(x) + 2xf'_k(x)]$, we have:

$$f_k(x) + 2xf'_k(x) > 0 \tag{9}$$

Since $h_k(1) = 1$, $h'_k(x) > 0$ and $h_k(x)$ is continuous, we have $\forall x \in (0, 1)$, $h_k(x) < 1$. We thus can obtain:

$$\begin{aligned}
f_{k+1}(x) &= \frac{1}{2}f_k(x)[3 - xf_k^2(x)] \\
&= \frac{1}{2}f_k(x)[3 - h_k(x)] \\
&> 0
\end{aligned} \tag{10}$$

Furthermore, $h'_{k+1}(x) = f_{k+1}(x)[f_{k+1}(x) + 2xf'_{k+1}(x)]$, where

$$\begin{aligned}
&f_{k+1}(x) + 2xf'_{k+1}(x) \\
=&\frac{3}{2}[f_k(x) + 2xf'_k(x)] - \frac{3}{2}xf_k^3(x) - 3x^2f_k^2(x)f'_k(x) \\
=&\frac{3}{2}[f_k(x) + 2xf'_k(x)] - \frac{3}{2}xf_k^2(x)[f_k(x) + 2xf'_k(x)] \\
=&\frac{3}{2}[1 - xf_k^2(x)][f_k(x) + 2xf'_k(x)] \\
=&\frac{3}{2}[1 - h_k(x)][f_k(x) + 2xf'_k(x)]
\end{aligned}$$

So we have $h'_{k+1}(x) = \frac{3}{2}f_{k+1}(x)[1 - h_k(x)][f_k(x) + 2xf'_k(x)] > 0$. Combining the result in Eqn. 10, we thus have it holds when $T = k + 1$.

As a result, we have $\forall T \in \mathbb{N}$, $f_T(x) > 0$ and $h'_T(x) > 0$, when $x \in (0, 1)$.

Since $h_T(1) = 1$ and $h_T(x)$ is continuous, we have $h_T(x) < 1$. Besides, we have $h_T(x) = x{f_T}^2(x) > 0$, then $h_T(x) \in (0, 1)$. $\qquad \square$

### A.2  PROOF OF PROPOSITION.2

**Proposition 2.**  Given $x \in (0, 1)$, $\forall T \in \mathbb{N}$, we have $h_{T+1}(x) > h_T(x)$.

*Proof.* According to proof of Proposition.1, we have that when $x \in (0, 1)$ and $\forall T \in \mathbb{N}$, $f_T(x) > 0$ and $h_T(x) = x{f_T}^2(x) \in (0, 1)$.

Therefore, we have $h_{T+1}(x) > h_T(x) \iff f_{T+1}(x) > f_T(x)$. It is obvious that

$$
\begin{aligned}
f_{k+1}(x) - f_k(x) &= \frac{3}{2} f_k(x) - \frac{1}{2} x f_k^3(x) - f_k(x) \\
&= \frac{1}{2} f_k(x) - \frac{1}{2} x f_k^3(x) \\
&= \frac{1}{2} f_k(x)[1 - x f_k^2(x)] \\
&= \frac{1}{2} f_k(x)[1 - h_k(x)] \\
&> 0
\end{aligned}
$$

So given $x \in (0, 1), \forall T \in \mathbb{N}$, we have $h_{T+1}(x) > h_T(x)$. $\qquad\square$

## B  PROOFS OF THEOREM

### B.1  PROOF OF THEOREM 1.

**Theorem 1.**  Define one-variable iterative function $f_T(x)$, satisfying

$$
f_{k+1}(x) = \tfrac{3}{2} f_k(x) - \tfrac{1}{2} x f_k{}^3(x), k \ge 0; f_0(x) = 1.
$$

The mapping function of IterNorm is

$$
g(\lambda) = f_T(\tfrac{\lambda}{tr(\Sigma)})/\sqrt{tr(\Sigma)},
$$

so that $\forall \lambda_i \in \lambda(\mathbf{Z})$, IterNorm maps it to $\widehat{\lambda}_i = \frac{\lambda_i}{tr(\Sigma)} f_T{}^2(\frac{\lambda_i}{tr(\Sigma)})$.

*Proof.*  Given $\Sigma = \mathbf{U}\Lambda\mathbf{U}^T$, $\Lambda = \mathrm{diag}(\lambda_1, \dots, \lambda_d)$, $\mathbf{U} = [\mathbf{u}_1, \dots, \mathbf{u}_d]$. Following the calculation steps of IterNorm, we have

$$
\Sigma_N = \Sigma/tr(\Sigma) = \sum_{i=1}^{d} \frac{\lambda_i}{tr(\Sigma)} \mathbf{u}_i \mathbf{u}_i{}^T \tag{11}
$$

Define

$$
\Phi'_T = \sum_{i=1}^{d} \frac{1}{\sqrt{tr(\Sigma)}} f_T(\frac{\lambda_i}{tr(\Sigma)}) \mathbf{u}_i \mathbf{u}_i{}^T \tag{12}
$$

Based on $\Phi_T = \sum_{i=1}^{d} g(\lambda_i) \mathbf{u}_i \mathbf{u}_i{}^T$, if we can prove $\Phi'_T = \Phi_T$, we will have

$$
g(\lambda) = \frac{1}{\sqrt{tr(\Sigma)}} f_T(\frac{\lambda}{tr(\Sigma)})
$$

Define $\mathbf{P}'_T = \sqrt{tr(\Sigma)}\Phi'_T$, then we have $\Phi'_T = \Phi_T \iff \mathbf{P}'_T = \mathbf{P}_T$. We can prove $\mathbf{P}'_T = \mathbf{P}_T$ by mathematical induction.
(1) When $T = 0$,

$$
f_0(\tfrac{\lambda_i}{tr(\Sigma)}) = 1, \mathbf{P}'_0 = \mathbf{P}_0 = \mathbf{I}
$$

(2) When $T \ge 1$, assume that $\mathbf{P}'_{T-1} = \mathbf{P}_{T-1}$, thus

$$
\begin{aligned}
\mathbf{P}_T &= \frac{3}{2}\mathbf{P}_{T-1} - \frac{1}{2}\mathbf{P}_{T-1}^3 \Sigma_N \\
&= \frac{3}{2}\mathbf{P}'_{T-1} - \frac{1}{2}(\mathbf{P}'_{T-1})^3 \Sigma_N
\end{aligned}
$$

According to the definition of $\mathbf{P}'_T$,

$$\mathbf{P}'_{T-1} = \sum_{i=1}^{d} f_{T-1}\left(\frac{\lambda_i}{tr(\Sigma)}\right)\mathbf{u}_i\mathbf{u}_i^T$$

Because $\forall i, \mathbf{u}_i^T\mathbf{u}_i = 1$ and $\forall i \neq j, \mathbf{u}_i^T\mathbf{u}_j = 0$,

$$\begin{aligned}
\mathbf{P}^3_{T-1}\Sigma_N &= (\mathbf{P}'_{T-1})^3\Sigma_N \\
&= \left(\sum_{i=1}^{d} f_{T-1}\left(\frac{\lambda_i}{tr(\Sigma)}\right)\mathbf{u}_i\mathbf{u}_i^T\right)^3\left(\sum_{i=1}^{d}\frac{\lambda_i}{tr(\Sigma)}\mathbf{u}_i\mathbf{u}_i^T\right) \\
&= \sum_{i=1}^{d} f^3_{T-1}\left(\frac{\lambda_i}{tr(\Sigma)}\right)\frac{\lambda_i}{tr(\Sigma)}\mathbf{u}_i\mathbf{u}_i^T
\end{aligned}$$

Therefore, we have

$$\begin{aligned}
\mathbf{P}_T &= \frac{3}{2}\mathbf{P}'_{T-1} - \frac{1}{2}(\mathbf{P}'_{T-1})^3\Sigma_N \\
&= \frac{3}{2}\sum_{i=1}^{d} f_{T-1}\left(\frac{\lambda_i}{tr(\Sigma)}\right)\mathbf{u}_i\mathbf{u}_i^T - \frac{1}{2}\sum_{i=1}^{d} f^3_{T-1}\left(\frac{\lambda_i}{tr(\Sigma)}\right)\frac{\lambda_i}{tr(\Sigma)}\mathbf{u}_i\mathbf{u}_i^T \\
&= \sum_{i=1}^{d}\left\{\frac{3}{2}f_{T-1}\left(\frac{\lambda_i}{tr(\Sigma)}\right) - \frac{1}{2}f^3_{T-1}\left(\frac{\lambda_i}{tr(\Sigma)}\right)\frac{\lambda_i}{tr(\Sigma)}\right\}\mathbf{u}_i\mathbf{u}_i^T
\end{aligned}$$

Note

$$f_T\left(\frac{\lambda_i}{tr(\Sigma)}\right) = \frac{3}{2}f_{T-1}\left(\frac{\lambda_i}{tr(\Sigma)}\right) - \frac{1}{2}f^3_{T-1}\left(\frac{\lambda_i}{tr(\Sigma)}\right)\frac{\lambda_i}{tr(\Sigma)}$$

So that

$$\mathbf{P}_T = \sum_{i=1}^{d} f_T\left(\frac{\lambda_i}{tr(\Sigma)}\right)\mathbf{u}_i\mathbf{u}_i^T = \mathbf{P}'_T$$

We obtain that

$$\Phi_T = \Phi'_T = \sum_{i=1}^{d}\frac{1}{\sqrt{tr(\Sigma)}}f_T\left(\frac{\lambda_i}{tr(\Sigma)}\right)\mathbf{u}_i\mathbf{u}_i^T = \mathbf{U}\frac{1}{\sqrt{tr(\Sigma)}}f_T\left(\frac{\Lambda}{tr(\Sigma)}\right)\mathbf{U}^T$$

Thus, the mapping function of IterNorm is $g(\lambda) = f_T\left(\frac{\lambda}{tr(\Sigma)}\right)/\sqrt{tr(\Sigma)}$. The whitened output is $\widehat{\mathbf{Z}} = \Phi_T\mathbf{Z}_c = \mathbf{U}\frac{1}{\sqrt{tr(\Sigma)}}f_T\left(\frac{\Lambda}{tr(\Sigma)}\right)\mathbf{U}^T\mathbf{Z}_c$. The covariance matrix of $\widehat{\mathbf{Z}}$ is

$$\Sigma_{\widehat{\mathbf{Z}}} = \frac{1}{m}\widehat{\mathbf{Z}}\widehat{\mathbf{Z}}^T = \mathbf{U}\frac{\Lambda}{tr(\Sigma)}f_T^2\left(\frac{\Lambda}{tr(\Sigma)}\right)\mathbf{U}^T = \sum_{i=1}^{d}\frac{\lambda_i}{tr(\Sigma)}f_T^2\left(\frac{\lambda_i}{tr(\Sigma)}\right)\mathbf{u}_i\mathbf{u}_i^T$$

So that $\forall\lambda_i \in \lambda(\mathbf{Z})$, IterNorm maps it to $\widehat{\lambda}_i = \frac{\lambda_i}{tr(\Sigma)}f_T^2\left(\frac{\lambda_i}{tr(\Sigma)}\right)$ which is a special instance of Spectral Transformation. □

### B.2 PROOF OF THEOREM 2.

**Theorem 2.** Let $\mathbf{x} \in [0,1]^d, \forall T \in \mathbb{N}_+$, $INTL(\mathbf{x})$ shown in Eqn. 7 is a strictly convex function. $\mathbf{x}^* = [\frac{1}{d}, \cdots, \frac{1}{d}]^T$ is the unique minimum point as well as the optimal solution to $INTL(\mathbf{x})$.

*Proof.* The INTL can be viewed as the following optimization problem:

$$\min_{\theta\in\Theta} \quad INTL(\mathbf{Z}) = \sum_{j=1}^{d}(1 - (\Sigma_{\widehat{\mathbf{z}}})_{jj})^2 \tag{13}$$

where $\mathbf{Z} = F_\theta(\cdot)$ and $\widehat{\mathbf{Z}} = IterNorm(\mathbf{Z})$. Eqn. 6 can be viewed as a optimization problem over $\theta$ to encourage the trace of $\widehat{\mathbf{Z}}$ to be $d$.

Let $(x_1, \cdots, x_d) = \varphi(\mathbf{Z})$, where $x_i = \lambda_i/tr(\Sigma)$ as defined in the submitted paper. If $\mathbf{Z} \in \mathbb{R}^{d \times m}$, $\varphi(\cdot)$ will be surjective from $\mathbb{R}^{d \times m}$ to $\mathbb{D}_\mathbf{x} = \{\mathbf{x} \in [0, 1]^d : x_1 + \cdots + x_d = 1\}$. When the range of $F_\theta(\cdot)$ is wide enough, for example, $F_\theta(\cdot)$ is surjective from $\theta \in \Theta$ to $\mathbf{Z} \in \mathbb{R}^{d \times m}$. Here we can view $F_\theta(\cdot)$ as a function over $\theta$, since the input is given and fixed. Then $\varphi(F_\theta(\cdot))$ is surjective from $\theta \in \Theta$ to $\mathbf{x} \in \mathbb{D}_\mathbf{x}$, meaning that if we find the optimal solution $\mathbf{x}^*$, we are able to get the corresponding $\theta^* \in \Theta$, subject to $\mathbf{x}^* = \varphi(F_{\theta^*}(\cdot))$. On the contrary, for any $\theta \in \Theta$, we can get $\mathbf{x} = \varphi(F_\theta(\cdot)) \in \mathbb{D}_\mathbf{x}$.

Therefore, the optimization expression for minimizing INTL can be written as follows which have the same range and optimal value as Eqn. 6:

$$
(P_{INTL}) \begin{cases} \min & INTL(\mathbf{x}) = \sum_{j=1}^{d} \left( \sum_{i=1}^{d} [1 - h_T(x_i)] u_{ji}^2 \right)^2 \\ s.t. & \sum_{i=1}^{d} x_i = 1 \\ & x_i \geq 0, i = 1, \cdots, d \end{cases}
\tag{14}
$$

We denote the Lagrange function of $P_{INTL}$ is that

$$
L(\mathbf{x}; \boldsymbol{\alpha}, \mu) = INTL(\mathbf{x}) + \sum_{i=1}^{d} \alpha_i(-x_i) + \mu \left( \sum_{i=1}^{d} x_i - 1 \right)
$$

### B.2.1 CONVEXITY AND CONCAVITY OF $h_T(x)$

Before calculating extreme points of $P_{INTL}$, we first consider the convexity and concavity of $h_T(x)$ which is critical to proof.

When $T = 0$, we have $h_0(x) = x$, so $h_0''(x) = 0$.

(1) When $T = 1$, we have $h_1(x) = f_1^2(x) = \frac{9}{4}x - \frac{3}{2}x^2 + \frac{1}{4}x^3$, so $h_1''(x) = \frac{3}{2}(x - 2) < 0$.

(2) Assume that when $T = k$, $h_k''(x) < 0$ holds. We can easily get following propositions by derivation:

$$
f_{k+1}'(x) = \frac{3}{2}f_k'(x) - \frac{1}{2}f_k^3(x) - \frac{3}{2}x f_k^2(x) f_k'(x)
\tag{15}
$$

$$
f_{k+1}''(x) = \frac{3}{2}f_k''(x) - 3f_k^2(x)f_k'(x) - \frac{3}{2}x f_k^2(x)f_k''(x) - 3x f_k(x)[f_k'(x)]^2
\tag{16}
$$

$$
h_{k+1}''(x) = 4f_{k+1}(x)f_{k+1}'(x) + 2x[f_{k+1}'(x)]^2 + 2x f_{k+1}(x)f_{k+1}''(x)
\tag{17}
$$

For convenience in our calculation, let $a = f_k(x), b = f_k'(x), c = f_k''(x)$, and $h = h_k(x) = xa^2$.

We split Eqn. 17 into three parts and take Eqn. 15 and 16 into calculation:

$$
\begin{aligned}
4f_{k+1}(x)f_{k+1}'(x) &= 4(\frac{3}{2}a - \frac{1}{2}ah)(\frac{3}{2}b - \frac{1}{2}a^3 - \frac{3}{2}bh) \\
&= a(3 - h)(3b - a^3 - 3bh) \\
2x[f_{k+1}'(x)]^2 &= 2x(\frac{3}{2}b - \frac{1}{2}a^3 - \frac{3}{2}bh)^2 \\
&= \frac{1}{2}x(3b - a^3 - 3bh)^2 \\
2x f_{k+1}(x)f_{k+1}''(x) &= 2(\frac{3}{2}a - \frac{1}{2}ah)[\frac{3}{2}c(1 - h) - 3a^2 b - 3xab^2] \\
&= \frac{1}{2}ax(3 - h)[3c(1 - h) - 6a^2 b - 6xab^2]
\end{aligned}
$$

Considering to construct the form of $h_k''(x) = 2(2ab + xac + xb^2)$, we first calculate that

$$
\begin{aligned}
&4f_{k+1}(x)f_{k+1}'(x) + 2xf_{k+1}(x)f_{k+1}''(x) \\
=&\frac{1}{2}(3-h)[6ab - 2a^4 - 6abh + 3xac(1-h) - 6abh - 6xb^2h] \\
=&\frac{1}{2}(3-h)[3xac(1-h) + 6ab(1-h) + 3xb^2(1-h) \\
&- 3xb^2(1-h) - 2a^4 - 6abh - 6xb^2h] \\
=&\frac{3}{4}(3-h)(1-h)h_k''(x) - \frac{1}{2}(3-h)(3xb^2h + 3xb^2 + 2a^4 + 6abh)
\end{aligned}
$$

Then we calculate the left part

$$
\begin{aligned}
2x[f_{k+1}'(x)]^2 =&\frac{1}{2}x(3b - a^3 - 3bh)^2 \\
=&\frac{1}{2}(9xb^2 + xa^6 + 9xb^2h^2 - 6xa^3b - 18xb^2h + 6xa^3bh) \\
=&\frac{1}{2}(9xb^2 + a^4h + 9xb^2h^2 - 6abh - 18xb^2h + 6abh^2)
\end{aligned}
$$

For convenience, let

$$
\begin{aligned}
S =& -\frac{1}{2}(3-h)(3xb^2h + 3xb^2 + 2a^4 + 6abh) \\
=&\frac{1}{2}(3xb^2h^2 + 3xb^2h + 2a^4h + 6abh^2 - 9xb^2h - 9xb^2 - 6a^4 - 18abh)
\end{aligned}
$$

Then we have

$$
\begin{aligned}
2x[f_{k+1}'(x)]^2 + S =&\frac{1}{2}(3a^4h + 12xb^2h^2 - 24abh - 24xb^2h + 12abh^2 - 6a^4) \\
=&\frac{3}{2}(h-2)(a^4 + 4abh + 4xb^2h) \\
=&\frac{3}{2}(h-2)(a^4 + 4xa^3b + 4x^2a^2b^2) \\
=&\frac{3}{2}(h-2)(a^2 + 2xab)^2 \\
=&\frac{3}{2}[h_k(x) - 2][h_k'(x)]^2
\end{aligned}
$$

Here we obtain $h_{k+1}''(x) = \frac{3}{4}[3 - h_k(x)][1 - h_k(x)]h_k''(x) + \frac{3}{2}[h_k(x) - 2][h_k'(x)]^2$. Based on Lemma.1, we know $h_k(x) \in (0,1)$, so $h_{k+1}''(x) < 0$. Therefore, when $x \in (0,1)$, then $\forall T \in \mathbb{N}_+$, $h_T(x) = xf_T^2(x)$ is a strictly concave function that satisfies $h_T''(x) < 0$ and $h_0''(x) = 0$.

### B.2.2 OPTIMAL SOLUTION FOR THE LAGRANGE FUNCTION

Based on Section B.2.1, when $x \in (0,1)$, then $\forall T \in \mathbb{N}_+$, $h_T(x) = xf_T^2(x)$ is a strictly concave function that satisfies $h_T''(x) < 0$. So $1 - h_T(x)$ is a strictly convex function.

We discuss $g_j(x_1, \cdots, x_d) = \sum_{i=1}^{d}[1 - h_T(x_i)]u_{ji}^2$ first. Denote that the Hessen Matrix of $g_j(x_1, \cdots, x_d)$ about $\mathbf{x}$ is

$$
\nabla^2 g_j = \begin{bmatrix} -u_{j1}^2 h_T''(x_1) & & \\ & \ddots & \\ & & -u_{jd}^2 h_T''(x_d) \end{bmatrix}
$$

and the Hessen Matrix of $g_j^2(x_1, \cdots, x_d)$ about $\mathbf{x}$ is

$$
\nabla^2(g_j^2) = \nabla(2g_j \nabla g_j) = 2g_j \nabla^2 g_j + 2(\nabla g_j)(\nabla g_j)^T
$$

We denote that all eigenvalues of $(\nabla g_j)(\nabla g_j)^T$ are $(\nabla g_j)^T(\nabla g_j), 0, \cdots, 0$. All eigenvalues are non-negtive, denoting that $2(\nabla g_j)(\nabla g_j)^T$ is semi-positive.

Now we denote that the Hessen Matrix of $INTL(\mathbf{x})$ is

$$\nabla^2 INTL(\mathbf{x}) = \sum_{j=1}^{d} \nabla^2 (g_j^2)$$

$$= 2\sum_{j=1}^{d} (\nabla g_j)(\nabla g_j)^T + 2\sum_{j=1}^{d} g_j \nabla^2 g_j$$

where

$$2\sum_{j=1}^{d} g_j \nabla^2 g_j = 2 \begin{bmatrix} -\sum_{j=1}^{d} u_{j1}^2 h_T''(x_1) g_j & & \\ & \ddots & \\ & & -\sum_{j=1}^{d} u_{jd}^2 h_T''(x_d) g_j \end{bmatrix}$$

We denote that $h_T''(x_i) < 0, g_j > 0$, and $u_{ji}$ are not all zeros for a certain $i$ (since $\sum_{j=1}^{d} u_{ji}^2 = 1$).

Therefore, $-\sum_{j=1}^{d} u_{ji}^2 h_T''(x_i) g_j > 0$ and $2\sum_{j=1}^{d} g_j \nabla^2 g_j$ must be a positive matrix.

Since $2\sum_{j=1}^{d} (\nabla g_j)(\nabla g_j)^T$ is semi-positive, then we can denote that $\nabla^2 INTL$ is positive.

Therefore, $INTL(\mathbf{x})$ is strictly convex about $\mathbf{x}$ on $(0,1)^d$.

And for $INTL(\mathbf{x})$ is continuous, the minimum point on $[0,1]^d$ is the same as that on $(0,1)^d$.

While the constraints of $(P_{INTL})$ form a convex set, $(P_{INTL})$ must be a convex programming, which means that the KKT point of $(P_{INTL})$ is its unique extreme point, and the global minimum point in the same time.

We denote that the KKT conditions of $(P_{INTL})$ is that

$$\begin{cases} \frac{\partial L}{\partial x_i} = 0, & i = 1, \cdots, d \\ \alpha_i(-x_i) = 0, & i = 1, \cdots, d \\ \alpha_i \geq 0, & i = 1, \cdots, d \\ \sum_{i=1}^{d} x_i - 1 = 0 \end{cases}$$

We can identify one of the solutions to the KKT conditions is that

$$\begin{cases} \mathbf{x} = [\frac{1}{d}, \cdots, \frac{1}{d}]^T \\ \boldsymbol{\alpha} = \mathbf{0} \\ \mu = -2h_T'(\frac{1}{d})[h_T(\frac{1}{d}) - 1] \end{cases}$$

It is easy to identify the last three equations in KKT conditions. As for the first equation, for all $t = 1, \cdots, d$, we have

$$
\begin{aligned}
\frac{\partial L}{\partial x_t} &= 2h_T'(x_t) \sum_{i=1}^{d} \sum_{j=1}^{d} [h_T(x_i) - 1] u_{ji}^2 u_{jt}^2 - \alpha_i + \mu \\
&= 2h_T'(\frac{1}{d}) \sum_{i=1}^{d} \sum_{j=1}^{d} [h_T(\frac{1}{d}) - 1] u_{ji}^2 u_{jt}^2 + \mu \\
&= 2h_T'(\frac{1}{d}) \sum_{j=1}^{d} [h_T(\frac{1}{d}) - 1] \left( \sum_{i=1}^{d} u_{ji}^2 \right) u_{jt}^2 + \mu \\
&= 2h_T'(\frac{1}{d}) \sum_{j=1}^{d} [h_T(\frac{1}{d}) - 1] u_{jt}^2 + \mu \\
&= 2h_T'(\frac{1}{d}) [h_T(\frac{1}{d}) - 1] \left( \sum_{j=1}^{d} u_{jt}^2 \right) + \mu \\
&= 2h_T'(\frac{1}{d}) [h_T(\frac{1}{d}) - 1] + \mu \\
&= 0
\end{aligned}
$$

Therefore, $\mathbf{x}^* = [\frac{1}{d}, \cdots, \frac{1}{d}]^T$ is the optimal solution to $(P_{INTL})$. INTL promotes the equality of all eigenvalues in the optimization process, which provides a theoretical guarantee to avoid dimensional collapse. $\qquad \square$

## C  ALGORITHM OF INTL

The description of our paper is based on batch whitening (BW) (Ermolov et al., 2021; Hua et al., 2021), and it can extend similarly for channel whitening (CW) (Weng et al., 2022), where the covariance matrix of $\mathbf{Z}$ is calculated as $\Sigma = \frac{1}{d}\mathbf{Z}^T\mathbf{Z}$. We implement INTL based on CW, considering CW is more effective when the batch size $m$ is relatively small.

Given the centralized embedding of two positive pairs $\mathbf{Z}^{(v)}, \mathbf{Z}^{(v)} \in \mathbb{R}^{d \times m}$ and $v \in \{1, 2\}$, we use IterNorm to obtain the approximately whitened output $\widehat{\mathbf{Z}}^{(v)} = [\hat{\mathbf{z}}_1^{(v)}, \ldots, \hat{\mathbf{z}}_m^{(v)}]$. The loss functions used in our method are

$$
\mathcal{L}_{MSE} = \frac{1}{m} \sum_i \| \frac{\hat{\mathbf{z}}_i^{(1)}}{\|\hat{\mathbf{z}}_i^{(1)}\|_2} - \frac{\hat{\mathbf{z}}_i^{(2)}}{\|\hat{\mathbf{z}}_i^{(2)}\|_2} \|_2^2 \tag{18}
$$

$$
\mathcal{L}_{trace} = \sum_{v=1}^{2} \sum_{i=1}^{m} (1 - \frac{1}{d} \hat{\mathbf{z}}_i^{(v)^T} \hat{\mathbf{z}}_i^{(v)})^2 \tag{19}
$$

$$
\mathcal{L}_{INTL} = \mathcal{L}_{MSE} + \beta \cdot \mathcal{L}_{trace} \tag{20}
$$

where $\mathcal{L}_{MSE}$ indicates MSE of $L_2-$normalized vectors which minimizes the distance between $\widehat{\mathbf{Z}}^{(1)}$ and $\widehat{\mathbf{Z}}^{(2)}$. Here we simplify the expression of $\mathcal{L}_{trace}$ in Eqn. 6, because off-diagonal elements of $\Sigma_{\hat{\mathbf{z}}}$ does not need to be calculated. $\beta$ is the trade-off between $\mathcal{L}_{MSE}$ and INTL.

In our experiments, we observe that when the iteration number $T$ of IterNorm is fixed, the coefficient $\beta$ that obtains good performance has only relevant to the batch size. So we fix the iteration number $T$ to 4 and empirically regress $\beta$ with various batch sizes and obtain that $\beta = 0.01 * (log_2 bs - 3)$ where $bs$ means the batch size and $bs > 8$. We keep the iteration number $T$ of IterNorm and the coefficient $\beta$ fixed in this form (i.e., $\beta$ is determined given the batch size) across all the datasets and architectures, so our INTL can be directly applied to other datasets and models without tuning the coefficient.

For clarity, we also describe the algorithm of INTL in PyTorch-style pseudocode, shown in Figure 4(a).

```
# f: backbone + projection
# bs: batch size
# aug: random augmentation

for x in loader:  # load a minibatch x with m samples
    z1, z2 = f(aug(x)), f(aug(x))  # embedding
    # transformed output
    z1_hat, z2_hat = IterNorm(z1), IterNorm(z2)
    # trade_off between MSE and trace Loss
    trade_off = (log2(bs) - 3) * 0.01
    mse = norm_mse(z1_hat, z2_hat)   # MSE
    trace_loss = TL(z1_hat) + TL(z2_hat)   # trace Loss
    loss = mse + trade_off * trace_loss
    return loss

def IterNorm(x, iters=4):  # Iterative Normalization
    M, D = x.size()  # x: m * d
    x = x - x.mean(dim=1).reshape(M, 1)
    sigma = (x @ x.T) / (D - 1)  # covariance matrix
    trace = sigma.diagonal().sum()
    sigma_norm = sigma / trace  # normalize sigma
    P = eye(M)  # identity matrix: m * m
    for _ in range(iters):
        P = 1/2 * (3 * P - matrix_power(P, 3) @ sigma_norm)
    return P / trace.sqrt() @ x

def TL(x):  #Trace Loss
    _, D = x.size()
    d = torch.pow(x, 2).sum(axis = 1) / (D - 1)
    tl = d.add_(-1).pow_(2).sum()
    return tl

def norm_mse(x0, x1):
    x0 = normalize(x0)  # L2-normalize
    x1 = normalize(x1)  # L2-normalize
    return 2 - 2 * (x0 * x1).sum(dim=-1).mean()
```

Figure 4: Algorithm of INTL, PyTorch-style Pseudocode.

# D   ANALYTICAL EXPERIMENTS

## D.1   EXPERIMENTS ON SYNTHETIC 2D DATASET

In section 3.2 of the submitted paper, we conduct experiments on the 2D dataset and report the results on with varying $p$. Here, we provide the details of the experimental setup, and further show the results of IterNorm (Huang et al., 2019) for SSL in this 2D dataset.

### D.1.1   DETAILS OF EXPERIMENTAL SETUPS

We synthesize a two-dimensional dataset with isotropic Gaussian blobs containing 512 sample points as shown in Figure 5(a). We construct a toy Siamese network (a simple three-layer neural network, including three fully connected (FC) layers, with BN and ReLU appended to the first two) as the encoder for this dataset. The dimensions of the network are $(2 - 16) - (16 - 16) - (16 - 2)$ that each bracket represents the input and output dimensions of each FC layer respectively. We then use MSE as the loss function and do not normalize the features before calculating the loss function.

We train the model by randomly shuffling the data into mini-batches, and set the batch size to 32. We use the stochastic gradient descent (SGD) algorithm with a learning rate of $0.1$. In terms of the data transformation, we only apply Gaussian noise as data augmentation and generate 2 views from each

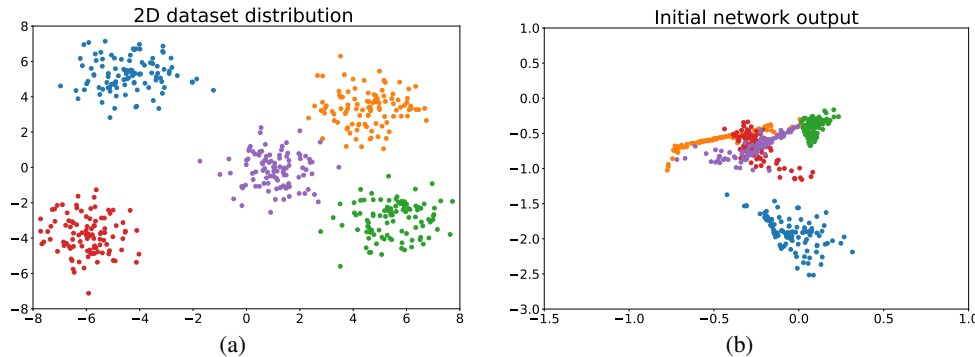

Figure 5: Visualization of our synthetic 2D dataset. We show (a) the distribution of our 2D dataset; (b) the initial output of the toy Siamese network.

sample point in mini-batches. We visualize the output of the initialized network without training in Figure 5(b). All runs are performed under the same random seed.

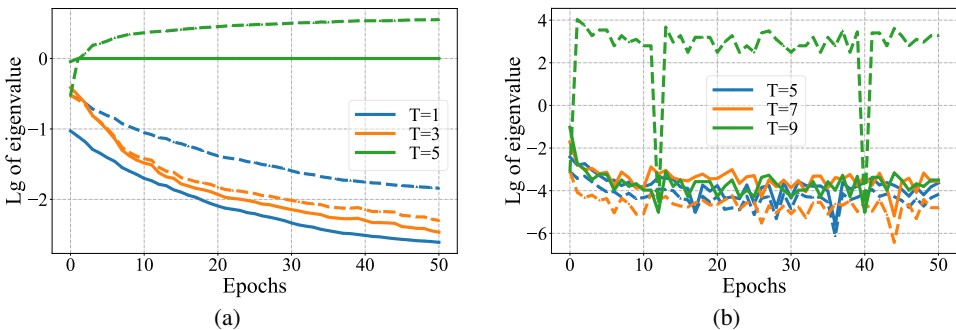

Figure 6: Investigate the spectrum of transformed output $\widehat{\mathbf{Z}}$ (solid lines) and the corresponding embedding $\mathbf{Z}$ (dashed lines) using IterNorm for SSL with different iteration numbers $T$. We show the evolution of eigenvalues during training on the toy 2D dataset (Note that there are only two eigenvalues and we ignore the larger one because it always remains a high value during training). In particular, (a) shows the results with a well-conditioned initial spectrum while (b) with a ill-conditioned one.

### D.1.2 RESULTS OF ITERNORM FOR SSL

To figure out the failure of IterNorm (Huang et al., 2019) for SSL, we further conduct experiments to investigate the spectrum of the whitened output $\widehat{\mathbf{Z}}$ using IterNorm on this synthetic 2D dataset for intuitive analyses. The output dimension of the toy model is 2, so there are only two eigenvalues of the covariance matrix of the output. We then track alterations of the two eigenvalues during training. IterNorm can obtain an idealized whitened output with a small iteration number (*e.g.*,T=5, as recommend in (Huang et al., 2019)) and avoid collapse, if the embedding $\mathbf{Z}$ has a well-conditioned spectrum[3] (Figure 6(a)). However, if the embedding $\mathbf{Z}$ has a ill-conditioned spectrum as shown in Figure 6(b), IterNorm fails to pull the small eigenvalue to approach 1 which results in dimensional collapse.

### D.2 EXPERIMENTS ON CIFAR-10

In section 3 and 4 of the submitted paper, we conduct several experiments on CIFAR-10 to illustrate our analysis. We provide a brief description of the setup in the caption of Figure 1 and 2 of the submitted paper. Here, we describe the details of these experiments. All experiments are uniformly based on the following training settings, unless otherwise stated in the figures of the submitted paper.

---

[3]A well-conditioned spectrum means that the condition number $c = \frac{\lambda_1}{\lambda_d}$ is small. Note $\lambda_1$ is the maximum eigenvalue and $\lambda_d$ is the minimum one.

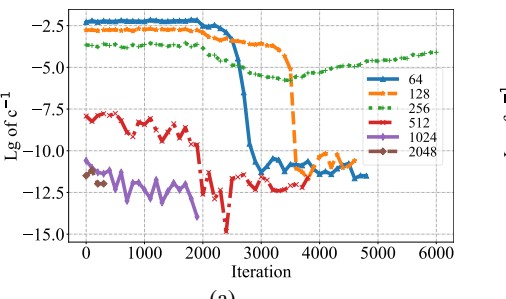 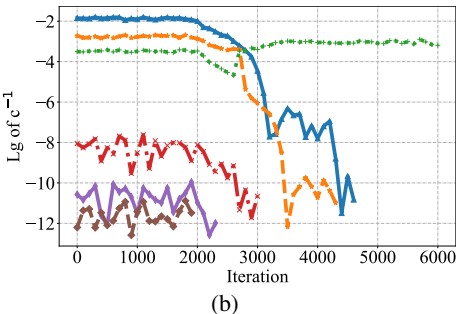

(a)                                        (b)

Figure 7: Investigate numerical instability of spectral transformation using power functions for SSL. The numbers in the legend represent embedding dimensions and the batch size is fixed to 512. (a) trains models on ImageNet with ResNet-50; (b) trains models on CIFAR-10 with ResNet-18; The models are trained for 6000 iterations, and we track the inverse of condition number ($c^{-1} = \frac{\lambda_d}{\lambda_1}$) in logarithmic scale with base 10 to judge whether the the covariance matrix is ill-conditioned. The models that were interrupted before the end of the training indicate training crash caused by numerical instability.

**Training Settings.**    We use the ResNet-18 as the encoder (the dimension of *encoding* is 512.), a two layer MLP with ReLU and BN appended as the projector (the dimension of the hidden layer and embedding are 1024 and 128 respectively). The model is trained on CIFAR-10 with a batch size of 256, using Adam optimizer (Kingma & Ba, 2014) with a learning rate of $3 \times 10^{-3}$, and learning rate warm-up for the first 500 iterations and a 0.2 learning rate drop at the last 50 and 25 epochs. The weight decay is set as $10^{-6}$. All transformations are performed with 2 positives extracted per image with standard data argumentation (see Section E.3 for details). We use the same evaluation protocol as in *W-MSE* (Ermolov et al., 2021).

**Method Settings.**    We use MSE of $L_2-$normalized vectors to be the loss function in all experiments. Specifically, in Figure 3 of the paper for the experiments of training the models with INTL, we simply set the trade-off parameter $\beta$ between MSE and INTL as follows: $\beta = 0.05$ for $T = 5$, $\beta = 0.5$ for $T = 3$ and $\beta = 5$ for $T = 1$ without fine-tuning. The details of INTL algorithm please refer to Section C.

### D.3    NUMERICAL INSTABILITY OF SPECTRAL TRANSFORMATION USING POWER FUNCTIONS

One issue with employing the spectral transformation $g(\lambda) = \lambda^{-p}$ (where $p$ is approximately 0.5) is the risk of numerical instability during the calculation of eigenvalues $\lambda$ and eigenvectors $\mathbf{U}$ via eigen-decomposition. This instability can arise when dealing with an ill-conditioned covariance matrix, as noted in (Paszke et al., 2019). In this study, we empirically validate the existence of this phenomenon in the context of self-supervised pre-training. It's important to mention that we primarily focus on the special case of $p = 0.5$, referred to as hard whitening, as similar phenomena are observed when $p$ is set near 0.5.

To assess the generality of this phenomenon, we conduct experiments on both ImageNet with ResNet-50 and CIFAR-10 with ResNet-18. We maintain a fixed batch size of 512 and manipulate the shape of the covariance matrix by adjusting the embedding dimension $d$ (where the covariance matrix has a shape of d × d). The models undergo 6000 iterations, and we monitor the inverse of the condition number ($c^{-1} = \frac{\lambda_d}{\lambda_1}$) to ascertain the ill-conditioned nature of the covariance matrix. The experimental results, depicted in Figure 7, lead to the following key observations:

(a) Training crashes when the embedding dimension exceeds the batch size (e.g., $d = 1024$ or 2048). In such cases, the covariance matrix becomes theoretically singular, and computing the inverse of the eigenvalues introduces numerical errors. However, in practice, the minimum eigenvalue of the covariance matrix is likely a very small non-zero value due to precision rounding or the use of a small constant. Consequently, the covariance matrix may already be ill-conditioned from the start of training. Both Figure 7(a) and (b) illustrate that when $d = 1024$ or 2048, the inverse of the condition number is approximately $10^{-12} \sim 10^{-10}$, indicating severe ill-conditioning from the beginning, resulting in rapid training breakdown.

Table 4: Parameters used for image augmentations on ImageNet and ImageNet-100.

| Parameter | $T_1$ | $T_2$ |
|---|---|---|
| crop size | $224 \times 224$ | $224 \times 224$ |
| maximum scale of crops | 1.0 | 1.0 |
| minimum scale of crops | 0.08 | 0.08 |
| brightness | 0.4 | 0.4 |
| contrast | 0.4 | 0.4 |
| saturation | 0.2 | 0.2 |
| hue | 0.1 | 0.1 |
| color jitter prob | 0.8 | 0.8 |
| horizontal flip prob | 0.5 | 0.5 |
| gaussian prob | 1.0 | 0.1 |
| solarization prob | 0.0 | 0.2 |

(b) Training is prone to crashing when the embedding dimension equals the batch size ($d = 512$). In such cases, it's challenging to definitively establish whether the covariance matrix is singular. However, our observations from Figure 7 suggest that the covariance matrix tends towards ill-conditioning when $d = 512$. The inverse of the condition number progressively decreases during training, eventually leading to training instability.

(c) There is a possibility of training instability when the embedding dimension is less than the batch size. In these situations, we initially observe that the covariance matrix remains well-conditioned. However, this favorable condition is not consistently maintained throughout training. We notice that well-conditioning suddenly breaks after a few iterations, leading to model collapse for $d = 64$ or $d = 128$. Interestingly, training does not crash when $d = 256$. This phenomenon was briefly discussed in (Ermolov et al., 2021), suggesting that stability can be improved by setting $m = 2d$.

We confirm the presence of numerical instability when employing hard whitening (Ermolov et al., 2021), as indicated by the above analysis. While one can mitigate this instability empirically by setting $m = 2d$, our experiments reveal that training crashes due to numerical instability can still occur at various points during training. In our extensive experimentation (with 10 random seeds and longer training iterations), we observed instances of numerical issues—approximately 3-4 times—occurring at different stages, including early, mid, or even towards the end of training. Even though it is possible to resume training using saved checkpoints in the event of a crash, this significantly limits the practical applicability of long-term pre-training.

# E    DETAILS OF EXPERIMENTS ON STANDARD SSL BENCHMARK

In this section, we provide the details of implementation and training protocol for the experiments on large-scale ImageNet (Deng et al., 2009), medium-scale ImageNet-100 (Tian et al., 2020a) and small-scale CIFAR-10/100 (Krizhevsky, 2009) classification as well as transfer learning to COCO (Lin et al., 2014) object detection and instance segmentation. We also provide computational overhead of INTL pre-training on ImageNet.

## E.1    DATASETS

• CIFAR-10 and CIFAR-100 (Krizhevsky, 2009), two small-scale datasets composed of $32 \times 32$ images with 10 and 100 classes, respectively.

• ImageNet-100 (Tian et al., 2020a), a random 100-class subset of ImageNet (Deng et al., 2009).

• ImageNet (Deng et al., 2009), the well-known largescale dataset with about 1.3M training images and 50K test images, spanning over 1000 classes.

• COCO2017 (Lin et al., 2014), a large-scale object detection, segmentation, and captioning dataset with 330K images containing 1.5 million object instances.

## E.2    EXPERIMENT ON IMAGENET

In section 5.1 of the paper, we compare our INTL to the state-of-the-art SSL methods on large-scale ImageNet classification. Here, we describe the training details of these experiments.

Table 5: Parameters used for multi-crop of INTL on ImageNet.

| Parameter | $T_1$ | $T_2$ | $T_3$ | $T_4$ | $T_5$ | $T_6$ |
|---|---|---|---|---|---|---|
| crop size | $224 \times 224$ | $224 \times 224$ | $192 \times 192$ | $160 \times 160$ | $128 \times 128$ | $96 \times 96$ |
| maximum scale of crops | 1.0 | 1.0 | 0.857 | 0.714 | 0.571 | 0.429 |
| minimum scale of crops | 0.2 | 0.2 | 0.171 | 0.143 | 0.114 | 0.086 |
| brightness | 0.4 | 0.4 | 0.4 | 0.4 | 0.4 | 0.4 |
| contrast | 0.4 | 0.4 | 0.4 | 0.4 | 0.4 | 0.4 |
| saturation | 0.2 | 0.2 | 0.2 | 0.2 | 0.2 | 0.2 |
| hue | 0.1 | 0.1 | 0.1 | 0.1 | 0.1 | 0.1 |
| color jitter prob | 0.8 | 0.8 | 0.8 | 0.8 | 0.8 | 0.8 |
| horizontal flip prob | 0.5 | 0.5 | 0.5 | 0.5 | 0.5 | 0.5 |
| gaussian prob | 0.5 | 0.5 | 0.5 | 0.5 | 0.5 | 0.5 |
| solarization prob | 0.1 | 0.1 | 0.1 | 0.1 | 0.1 | 0.1 |

Table 6: Parameters used for INTL pre-training on ImageNet-100.

| Parameter | Value |
|---|---|
| max epoch | 400 |
| backbone | ResNet-18 |
| projection layers | 3 |
| projection hidden dimension | 4096 |
| projection output dimension | 4096 |
| optimizer | SGD |
| SGD momentum | 0.9 |
| learning rate | 0.5 |
| learning rate warm-up | 2 epochs |
| learning rate schedule | cosine decay |
| weight decay | 2.5e-5 |
| batch size | 128 |

**Backbone and Projection.** We use the ResNet-50 (He et al., 2016) as the backbone and the output dimension is 2048. We use a 3-layers MLP as the projection: two hidden layers with BN and ReLU applied to it and a linear layer as output. We set dimensions of the hidden layer and embedding to 8192 as our initial experiments followed the settings of VICReg and Barlow Twins, both of which use a dimension of 8192 for the projection. Compared to a projection dimension of 2048, using a projection dimension of 8192 can bring about a $0.14\%$ improvement in top-1 accuracy for INTL. Therefore, we followed this setting in subsequent experiments on ImageNet. We report that using a projection dimension of 8192 requires approximately $18\%$ additional GPU memory and $2\%$ time per epoch compared to using the one of 2048.

**Image Transformation Details.** In image transformation, We use the same augmentation parameters as BYOL (Grill et al., 2020). Each input image is transformed twice to produce the two distorted views. The image augmentation pipeline consists of the following transformations: random cropping, resizing to $224 \times 224$, horizontal flipping, color jittering, converting to grayscale, Gaussian blurring, and solarization. The details of parameters are shown in Table 4.

**Optimizer and Learning Rate Schedule.** We apply the SGD optimizer, using a learning rate of *base-lr* × BatchSize / 256 and cosine decay schedule. The *base-lr* for 100-epoch pre-training is 0.5, for 200(400)-epoch is 0.4 and for 800-epoch is 0.3. The weight decay is $10^{-5}$ and the SGD momentum is 0.9. In addition, we use learning rate warm-up for the first 2 epochs of the optimizer.

**Evaluation Protocol.** For linear classification, we train the *linear classifier* for 100 epochs with SGD optimizer (using a learning rate of *base-lr* × BatchSize / 256 with a *base-lr* of 0.2) and using *MultiStepLR* scheduler with $\gamma = 0.1$ dropping at the last 40 and 20 epochs. Note that when combining INTL with multi-crop in the ablation experiments, the *base-lr* is set to 0.4. The batch size and weight decay for both are 256 and 0 respectively.

**Exponential Moving Average.** In the main text, we observe that our INTL can performs even better when utilized in conjunction with the Exponential Moving Average (EMA) technique. We set the base coefficient for momentum updating to 0.996 for all-epoch training. The momentum coefficient follows a cosine increasing schedule with final value of 1.0 as BYOL (Grill et al., 2020).

Table 7: Parameters used for INTL pre-training on CIFAR-10/100.

| Parameter | Value |
|---|---|
| max epoch | 1000 |
| backbone | ResNet-18 |
| projection layers | 3 |
| projection hidden dimension | 2048 |
| projection output dimension | 2048 |
| optimizer | SGD |
| SGD momentum | 0.9 |
| learning rate | 0.3 |
| learning rate warm-up | 2 epochs |
| learning rate schedule | cosine decay |
| weight decay | 1e-4 |
| batch size | 256 |

### E.3 EXPERIMENTS FOR SMALL AND MEDIUM SIZE DATASETS

In section 5.1 of the paper, we provide the classification results of INTL pre-training on small and medium size datasets such as CIFAR-10, CIFAR-100 and ImageNet-100. Here, We describe the details of implementation and training protocol for the experiments on these datasets as follows. For fairness, most of hyper-parameters we used such as batch size, projection settings, data augmentation and so on are consistent with solo-learn (da Costa et al., 2022).

**Experimental setup on ImageNet-100.** Details of implementation and training protocol for INTL pre-training on ImageNet-100 are shown in Table 6. The image transformation and evaluation protocol are the same as ones on ImageNet.

**Experimental setup on CIFAR-10/100.** Then Details of implementation and training protocol for INTL pre-training on CIFAR-10/100 are shown in Table 7. The details of image transformation are shown in Table 8. For evaluation, we use the same setup of protocol as in *W-MSE* (Ermolov et al., 2021): training the linear classifier for 500 epochs using the Adam optimizer and the labeled training set of each specific dataset, without data augmentation; the learning rate is exponentially decayed from $10^{-2}$ to $10^{-6}$ and the weight decay is $5 \times 10^{-6}$.

In addition, we also evaluate the accuracy of a k-nearest neighbors classifier (k-NN, k = 5) in these experiments. For other methods, we evaluate the models provided by (da Costa et al., 2022) to obtain k-NN accuracy which does not require additional parameters and training.

### E.4 EXPERIMENTS FOR TRANSFER LEARNING

In this part, we describe the training details of experiments for transfer learning. Our implementation is based on the released codebase of MoCo (He et al., 2020) [4] for transfer learning to object detection and instance segmentation tasks. We use the default hyper-parameter configurations from the training scripts provided by the codebase for INTL, using our 200-epoch and 800-epoch pre-trained model on ImageNet.

For the experiments of *COCO detection and COCO instance segmentation*, we use Mask R-CNN (1× schedule) fine-tuned in COCO 2017 train, evaluated in COCO 2017 val. The Mask R-CNN model is with the C4-backbone. Our INTL is performed with 3 random seeds, with mean and standard deviation reported.

### E.5 COMPUTATIONAL OVERHEAD

In Table 9, we report compute and GPU memory requirements based on our implementation for different settings on ImageNet with ResNet-50. The batch size is 256, and we train each model with 2 A100-PCIE-40GB GPUs, using mixed precision and py-torch optimized version of synchronized batch-normalization layers.

---

[4] *https://github.com/facebookresearch/moco/tree/main/detection* under the CC-BY-NC 4.0 license.

Table 8: Parameters used for image augmentations on CIFAR-10/100.

| Parameter | $T_1$ | $T_2$ |
|---|---|---|
| crop size | $32 \times 32$ | $32 \times 32$ |
| maximum scale of crops | 1.0 | 1.0 |
| minimum scale of crops | 0.08 | 0.08 |
| brightness | 0.4 | 0.4 |
| contrast | 0.4 | 0.4 |
| saturation | 0.2 | 0.2 |
| hue | 0.1 | 0.1 |
| color jitter prob | 0.8 | 0.8 |
| horizontal flip prob | 0.5 | 0.5 |
| gaussian prob | 0 | 0 |
| solarization prob | 0.0 | 0.2 |

Table 9: Computational cost. We report time and GPU memory requirements of our implementation for INTL trained per epoch on ImageNet with ResNet-50.

| Method | EMA | Multi-Crop | time / 1 epoch | peak memory / GPU |
|---|---|---|---|---|
| | No | No | 29min11 | 16.0 G |
| **INTL** | Yes | No | 24min46 | 11.8 G |
| | No | Yes | 57min33 | 25.9 G |
| | Yes | Yes | 50min52 | 21.2 G |

## F  ABLATION STUDY

In this section, we conduct a comprehensive set of ablation experiments to assess the robustness and versatility of our INTL. These experiments cover various aspects, including batch sizes, embedding dimensions, the use of multi-crop augmentation, semi-supervised training, the choice of Vision Transformer (ViT) backbones and adding trace loss to other methods.

**Batch size.**  Most SSL methods, including certain whitening-based methods, are known to be sensitive to batch sizes, e.g. SimCLR (Chen et al., 2020a), SwAV (Caron et al., 2020) and W-MSE (Ermolov et al., 2021) all require a large batch size (e.g. 4096) to work well. We then test the robustness of INTL to batch sizes. We train INTL on ImageNet for 100 epochs with various batch sizes ranging from 32 to

Table 10: Effect of batch sizes for INTL. We train 100 epoch on ImageNet and provide the Top-1 accuracy using linear evaluation. The embedding dimension is fixed to 8192.

| Bs | 32 | 64 | 128 | 256 | 512 | 1024 |
|---|---|---|---|---|---|---|
| acc.(%) | 64.2 | 66.4 | 68.1 | 68.7 | 69.5 | 69.7 |

1024. As shown in Table. 10, even if the batch size is as low as 32 or 64, INTL still maintains good performance. At the same time, when the batch size increases, the accuracy of INTL is also improved. These results indicate that INTL has good robustness to batch sizes and can adapt to various scenarios that constrain the training batch size.

**Embedding dimension.**  Embedding dimension, the output dimension of the projection, is also a key element for most self-supervised learning methods, which may have a significant impact on training results. As illustrated in (Zbontar et al., 2021), Barlow Twins is very sensitive to embedding dimension and it requires a large dimension (e.g. 8192 or 16384) to work well. We also test the robustness of INTL to embedding dimensions. Following the setup of (Chen et al., 2020a) and (Zbontar et al., 2021), we train INTL on ImageNet for 300 epochs with the dimension ranging from 64 to 16384. As shown in Figure. 8, even when the embedding dimension is low as 64 or 128, INTL still achieves good results. These results show that INTL also has strong robustness to embedding dimensions.

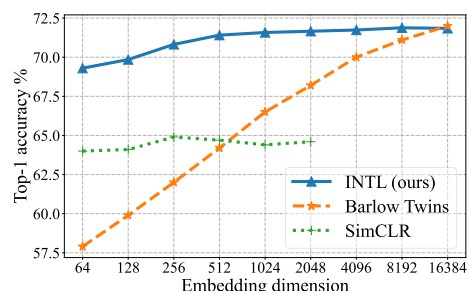

Figure 8: Ablation experiments for varying embedding dimensions. The batch size is fixed to 256.

**Multi-Crop.**  In the main text experiments, we employ the standard augmentation, which generates two augmented views for each sample. It's worth noting that multi-crop strategies, such as the

Table 11: Ablation experiments on ImageNet linear classification with EMA and multi-crop. All are based on ResNet-50 backbone.

| Method | Bs | EMA | Multi-Crop | 100 eps | 200 eps | 400 eps | 800 eps |
|---|---|---|---|---|---|---|---|
| SwAV | 4096 | No | No | 66.5 | 69.1 | 70.7 | 71.8 |
| **INTL (ours)** | 512 | No | No | **69.5** | **71.1** | **72.4** | **73.1** |
| SwAV | 4096 | No | Yes | 72.1 | 73.9 | 74.6 | 75.3 |
| SwAV | 256 | No | Yes | - | 72.7 | 74.3 | - |
| **INTL (ours)** | 256 | No | Yes | **72.4** | **74.3** | **74.9** | - |
| CLSA | 256 | Yes | No | - | 69.4 | - | 72.2 |
| **INTL (ours)** | 256 | Yes | No | 69.2 | **71.5** | 73.7 | **74.3** |
| DINO | 4080 | Yes | Yes | - | - | - | 75.3 |
| CLSA | 256 | Yes | Yes | - | 73.3 | - | 76.2 |
| **INTL (ours)** | 256 | Yes | Yes | 73.5 | **75.2** | 76.1 | **76.6** |

Table 12: Semi-supervised classification on top of the fine-tuned representations from 1% and 10% of ImageNet samples.

| Method | Epoch | Bs | Semi-supervised | | | |
|---|---|---|---|---|---|---|
| | | | Top-1 | | Top-5 | |
| | | | 1% | 10% | 1% | 10% |
| Supervised | 120 | 256 | 25.4 | 56.4 | 48.4 | 80.4 |
| SimCLR | 800 | 4096 | 48.3 | 65.6 | 75.5 | 87.8 |
| BYOL | 1000 | 4096 | 53.2 | 68.8 | 78.4 | 89.0 |
| SwAV | 800 | 4096 | 53.9 | 70.2 | 78.5 | 89.9 |
| Barlow Twins | 1000 | 2048 | **55.0** | **69.7** | 79.2 | 89.3 |
| VICReg | 1000 | 2048 | 54.8 | 69.5 | 79.4 | 89.5 |
| **INTL (ours)** | 800 | 512 | **55.0** | 69.4 | **80.8** | **89.8** |

one used by SwAV (Caron et al., 2020), are widely recognized for enhancing the performance of SSL methods. For instance, SwAV achieves a remarkable Top-1 accuracy of 75.3% with multi-crop. Therefore, we also conduct experiments with INTL using multi-crop. We apply an efficient multi-crop approach that generates 6 views for each image, with sizes of $2 \times 224 + 192 + 160 + 128 + 96$, which is similar to the approach used by CLSA (Wang & Qi, 2022). (Detailed parameter settings are provided in Table 5). The results are shown in Table 11. When INTL is paired with multi-crop augmentation, it consistently achieve notable improvements in top-1 accuracy. For instance, after 800 epochs of pre-training, INTL attains an impressive top-1 accuracy of 76.6%, even surpassing the common supervised baseline of 76.5%. The incorporation of multi-crop augmentation enhances the performance of INTL, making it a promising method for self-supervised representation learning across a range of experimental setups.

**Semi-supervised training.** For semi-supervised classification, we fine-tune our pre-trained INTL backbone and train the linear classifier on ImageNet for 20 epochs. We employ subsets of size 1% and 10%, following the same split as SimCLR. The optimization is performed using the SGD optimizer with a base learning rate of 0.006 for the backbone and 0.2 for the classifier, along with a cosine decay schedule. The semi-supervised results on the ImageNet validation dataset are presented in Table 12, demonstrating that INTL performs well in semi-supervised training scenarios.

**Vison Transformer backbones.** We conduct additional experiments using vision transformer (ViT) backbones for INTL. For comparison, we reproduce five other method under the same settings. The results are shown in Figure 9, illustrating that INTL maintains strong performance when ViTs are used as backbones. This suggests that INTL exhibits robust generalization capabilities across different network architectures.

**Barlow Twins/VICReg with Trace loss.** We conducte experiments on CIFAR-10/100 and ImageNet-100 to assess the impact of adding trace loss to Barlow Twins and VICReg, following the experimental setup outlined in Table 2 of our paper. We trained the models on CIFAR-10/100 for 200 epochs and on ImageNet-100 for 100 epochs. The coefficient of trace loss was set to 0.01, an empirically suitable value for both methods. The results are presented in the Table 13. We observed that adding trace loss to Barlow Twins had a minor positive effect on performance, while introducing it to VICReg significantly reduced performance, particularly on ImageNet-100. We hypothesize that this discrepancy may arise from the influence of trace loss on the regularization strength of these

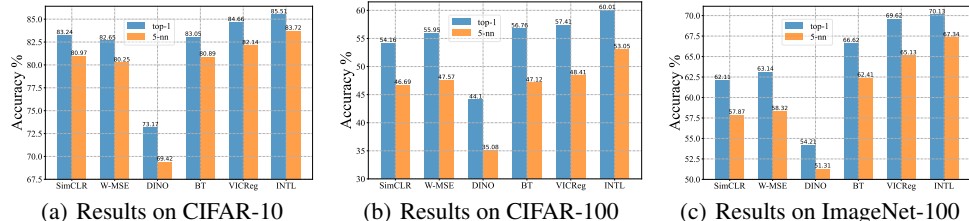

(a) Results on CIFAR-10     (b) Results on CIFAR-100     (c) Results on ImageNet-100

Figure 9: Abalation experiments using vision transformer (ViT) backbones. We train our INTL as well as other 5 methods (including SimCLR, W-MSE, DINO, Barlow Twins, and VICReg) for comparison when using ViTs as backbones. Our training setup involved ViT-tiny for 200 epochs on CIFAR-10/100 and ViT-small for 100 epochs on ImageNet-100. The settings were kept consistent with DINO, with the exception of the embedding dimension for W-MSE, which was set to 64, while other methods used 2048. We evaluated their classification performance using both a linear classifier and a 5-nearest neighbors classifier. The results for CIFAR-10, CIFAR-100, and ImageNet-100 are presented in panels (a), (b), and (c) respectively.

Table 13: Evaluate the performance of adding trace loss to BarlowTwins/VICReg.

| Method | CIFAR-10 | | CIFAR-100 | | ImageNet-100 | |
|---|---|---|---|---|---|---|
| | top-1 | 5-nn | top-1 | 5-nn | top-1 | 5-nn |
| Barlow Twins | 80.43 | **76.68** | 51.60 | 42.71 | 58.34 | 50.21 |
| Barlow Twins + trace loss | **80.45** | 76.32 | **51.66** | **43.94** | **59.78** | **50.45** |
| VICReg | **83.14** | **79.62** | **55.96** | **46.71** | **66.01** | **57.76** |
| VICReg + trace loss | 81.67 | 78.74 | 54.75 | 46.24 | 63.54 | 55.18 |

methods. It can either disrupt the existing balance, leading to reduced performance, or achieve a more favorable balance, resulting in improved performance.

# G  LICENSES OF DATASETS

ImageNet (Deng et al., 2009) is subject to the ImageNet terms of access: (contributors, 2020)

COCO (Lin et al., 2014). The annotations are under the Creative Commons Attribution 4.0 License. The images are subject to the Flickr terms of use (Flickr, 2020).

