# OpenReview forum: "Modulate Your Spectrum in Self-Supervised Learning"
_ICLR.cc/2024/Conference — ICLR 2024 poster_

### Official Review · Reviewer_UjJ7 · 2023-10-29

**Soundness:** 3 good
**Presentation:** 3 good
**Contribution:** 3 good
**Rating:** 6
**Confidence:** 4

**Summary:**

- The authors propose a framework referred to as Spectral Transformation (ST) to modulate the spectrum of embedding and to seek functions beyond whitening that can avoid dimensional collapse.
- Additionally, The authors introduce a novel ST instance named IterNorm with trace loss (INTL) to prevent collapse and modulate the spectrum of embedding toward equal eigenvalues during optimization.
- The extensive experiments on ImageNet classification and COCO object detection demonstrate the effectiveness of INTL in learning superior representations.

**Strengths:**

- (+) The authors show the novel points of the proposed methods, INTL, while comparing them with previous methods such as hard and soft whitening.
- (+) The authors show the empirical observations of IterNorm, which map all non-zero eigenvalues to approach one, with large enough iterations (T).

**Weaknesses:**

- (-) The authors seem to have a missing baseline [1] in SSL. The baseline looks similar to INTL from the viewpoint of spectral adjusting.
    - [1] Exploring the Gap between Collapsed & Whitened Features in SSL-ICML2022

**Questions:**

- Could the authors compare the INTL method with the baseline [1] if possible?

**Details Of Ethics Concerns:**

None.

---

> ### Author Response · Authors · 2023-11-14
> **Thank you sincerely for your valuable suggestions and insightful comments.**
>
> ## **Responses**
> Thanks sincerely for your encouraging and insightful comments. Please find our responses to specific concerns below.
>
> **Concern 1:** The authors seem to have a missing baseline [1] in SSL. The baseline looks similar to INTL from the viewpoint of spectral adjusting.
>
> **Response:**
> Thank you sincerely for your valuable suggestions. The theoretical contribution of [1] has been discussed in the related work of our submission, here we further list the key distinctions between our work and [1] as follows.
> 1. The fundamental concept in [1] revolves around the impact of feature whitening on generalization, with an emphasis on adjusting the whitening degree of pre-trained SSL models based on the power law behavior of eigenspectra to **enhance downstream performance**. In contrast, our paper seeks to introduce a novel SSL training algorithm, leveraging our ST framework, to **address dimensional collapse** without relying solely on whitening.
> 2. Our paper focuses on analyzing how adjusting the spectrum during the **training phase** can mitigate collapse-related issues. Conversely, [1] primarily investigates how the feature spectrum influences generalization during the **evaluation stage**.
> 3. The method proposed in [1], named PMP, serves as a **post-processing evaluation technique** aimed at adjusting the spectrum of a pre-trained SSL encoder for improved downstream performance. In contrast, our proposed method, INTL, aligns with **SSL training algorithms** like SimCLR, Barlow Twins, etc. It is designed to prevent collapse within joint embedding architectures during the training process.
>
> [1] demonstrates that the proposed PMP method is effective in enhancing the linear evaluation performance of encoders pre-trained by SimCLR and Barlow Twins, particularly on datasets with limited labeled samples. In line with your recommendations, we conducted experiments on 1% and 10% subsets of ImageNet to assess the performance of our INTL using PMP. The results, presented in the table below, include LP (standard linear probe) and PMP (the evaluation method proposed in [1]). The findings indicate that PMP contributes to the improved evaluation performance of INTL.
> &emsp; However, it's essential to note that our experiments in Section 5 are conducted based on the standard SSL benchmark using a standard linear probe, and therefore, the baseline from [1] is not incorporated in our experimental setup.
>
> |  Method   | Eval | Top-1 | Top-5  |
> | :-------- | :--------:| :--------:| :--------:|
> |    |  |1% &nbsp; 10%| 1% &nbsp; 10%|
> | SimCLR  | LP |48.1  &nbsp;   61.0 |  73.8   &nbsp;  84.3|
> | SimCLR  | PMP |50.9   &nbsp;  62.5 | 76.6  &nbsp;   85.2  |
>  |  |  |  |
> | INTL (ours) | LP  | 53.9   &nbsp;  66.4 | 79.1 &nbsp;   87.7  |
> | INTL (ours) | PMP  | 54.9  &nbsp;   67.8 | 79.6   &nbsp;   89.1  |

---

> > ### Author Response · Authors · 2023-11-22
> > **Official Comment by Authors**
> >
> > Dear Reviewer UjJ7,
> >
> > We recognize that the timing of this discussion period may not align perfectly with your schedule, yet we would greatly value the opportunity to continue our dialogue before the deadline approaches.
> >
> > Could you let us know if your concerns have been adequately addressed? If not, please feel free to raise them, and we are more than willing to provide further clarification; if you find that your concerns have been resolved, we would appreciate if you could re-consider the review score.
> >
> > We hope that we have resolved all your questions, but please let us know if there is anything more.
> >
> > Best wishes to you!

---

### Official Review · Reviewer_AFis · 2023-10-30

**Soundness:** 3 good
**Presentation:** 2 fair
**Contribution:** 2 fair
**Rating:** 6
**Confidence:** 3

**Summary:**

This paper proposed a new self-supervised learning framework called spectral transformation (ST) to modulate the spectrum of embedding to avoid dimensional collapse. To be specific, they introduced a novel ST instance named IterNorm with trace loss (INTL). Theoretically, this paper proved that INTL can modulate the spectrum of embeddings toward equal eigenvalues and prevent dimensional collapse. Empirically, the authors showed that INTL can obtain state-of-the-art performance for SSL on real-world datasets.

**Strengths:**

1. The dimensional collapse is an important problem in contrastive learning and the analysis in this paper is insightful.
2. The theoretical analysis and empirical results cooperate well. The improvements on real-world datasets are significant, especially in transfer learning tasks.

**Weaknesses:**

1. As analyzed in this paper, both whitening methods and INTL are instances of spectral transformations. However, it seems that INTL outperforms whitening in every task. So what are the disadvantages of whitening methods? It would be better to provide more theoretical and empirical comparisons between them.
2. The motivation behind the trace loss is a little confusing. Is it possible to provide a more detailed discussion?
3. It seems that INTL shows superior performance in 5-nn accuracy than linear probing accuracy. Are there any intuitive explanations for that?
4. There are some typos. For example, in p.4, ‘Eqn. 13 can be viewed as an optimization problem over …’ should be replaced with ‘Eqn.6 …’.

**Questions:**

see my comments above.

---

> ### Author Response · Authors · 2023-11-14
> **Thanks for your constructive comments and suggestions.**
>
> ## **Responses (1/2)**
>
> Your comments and suggestions are exceedingly helpful to improve our paper. Our point-to-point responses to your questions are given below.
>
> **Question 1:** It seems that INTL outperforms whitening in every task. So what are the disadvantages of whitening methods?
>
> **Response:** Following your constructive suggestions, we further analysis the disadvantages of whitening methods and explain why our INTL can outperform whitening methods in downstream tasks.
> 1. **The disadvantages of whitening methods.**
> >&emsp; In addition to the numerical instability issues discussed in Appendix D.3 for existing whitening methods, the unsatisfactory performance of these methods can be attributed to their weak constraints on embedding which are challenging to control and enhance effectively.
> >&emsp; Recent theoretical work [1] proposes a power law to analyze the relationship between eigenspectrum and representation quality. Meanwhile, [2] demonstrates that the degree of feature whitening influences generalization. Both of these works emphasize that **a steep spectrum diminishes representation quality, a smooth spectrum compromises generalization, while a moderate spectrum ensures optimal generalization ability**. SSL methods impose their constraints on embedding to prevent collapse, and the strength of these constraints directly influences the spectrum distribution.
> >&emsp; Hard whitening methods, such as W-MSE and CW-RGP, impose a full-rank constraint on embedding [3]. However, [3] indicates that the full-rank constraint is too weak to achieve optimal performance, leading to the proposal of a random group partition technique to enhance constraints and improve results. Nevertheless, this enhancement is unpredictable and comes with additional computational costs, especially on large datasets.
> >&emsp; Soft whitening methods, like Barlow Twins and VICReg, impose a strong whitening constraint on embedding, urging the covariance matrix of the embedding to be identity. The results in Table 3 show that these methods (Barlow Twins) perform even worse than hard whitening methods (W-MSE) in transfer learning, which indicates that excessive constraints have already compromised their generalization ability for downstream tasks.
>
> 2. **Why our INTL can outperform whitening methods in downstream tasks**
> >&emsp; The commendable performance of INTL can be attributed to its moderate constraint on the embedding. Unlike whitening methods, our INTL provides a moderate equal-eigenvalues constraint on embedding, which is stronger than hard whitening (full-rank constraint) but weaker than soft whitening (whitening constraint) as illustrated in Section 4. This moderate constraint results in a satisfactory spectrum distribution, thereby endowing the representation with better generalization capabilities.
>
> **Question 2:** The motivation behind the trace loss is a little confusing. Is it possible to provide a more detailed discussion?
>
> **Response:** As illustrated in Theorem 1, IterNorm implicitly maps  $\\forall \\lambda\_i \\in \\lambda(\\mathbf{Z})$ to $\\widehat{\\lambda}\_i = \\frac{\\lambda\_i}{tr(\\Sigma)} {f\_T}^2(\\frac{\\lambda\_i}{tr(\\Sigma)})$. Based on Formula 4 and 5, we know  $\\forall T\\in \\mathbb{N}, \\lambda\_i > 0 \\Longrightarrow 0 < \\widehat{\\lambda}\_i < 1 $, and $\\lim\_{T\\to\\infty} \\widehat{\\lambda}\_i = 1 $. This suggests that the eigenvalues of the covariance matrix $\\Sigma\_{\\widehat{\\mathbf{Z}}}$ resulting from the transformation output by IterNorm all fall within the range of 0 to 1, with an ideal convergence to 1 in the limit of infinite iterations ($T$). Therefore, $\\forall T\\in \\mathbb{N}, trace(\\Sigma\_{\\widehat{\\mathbf{Z}}})=\\sum\\limits\_{i=1}^d \\widehat{\\lambda}\_i < d$.
> &emsp; Consequently, in experiments, we have noted that IterNorm encounters severe dimensional collapse and struggles to effectively train the model in self-supervised learning, regardless of the chosen value for T. In such instances, we observe some eigenvalues of the covariance matrix $\\Sigma\_{\\widehat{\\mathbf{Z}}}$ collapse into 0, causing a substantial gap between $trace(\\Sigma\_{\\widehat{\\mathbf{Z}}})$ and $d$.
>
> &emsp; To address this issue, the motivation behind the trace loss is to introduce an additional penalty that encourages  $trace(\\Sigma\_{\\widehat{\\mathbf{Z}}})$ to approach its ideal maximum, where $\\sum\\limits\_{i=1}^d \\widehat{\\lambda}\_i = d$ and $\\forall i, \\widehat{\\lambda}\_i =1$ . By doing so, the collapse is mitigated since it becomes improbable for eigenvalues of embedding to be 0.

---

> > ### Author Response · Authors · 2023-11-14
> > **Thank you for your elaborate review.**
> >
> > ## **Responses (2/2)**
> > **Question 3:** It seems that INTL shows superior performance in 5-nn accuracy than linear probing accuracy. Are there any intuitive explanations for that?
> >
> > **Response:** Thanks for your inspiring questions! Our intuitive explanation is as follows:
> > &emsp; The accuracy of the k-nn classifier indicates the state of the sample distribution within the representation's local structure. The k-nn classifier tends to exhibit better classification performance under the condition that samples of the same class are closer in the feature space and those of different classes are farther apart. However, under the absence of data labels in SSL, ensuring this condition can be challenging. For instance, although contrastive learning aims to pull positive pairs together and push negative pairs apart from positive one, the negative pairs may belong to the same class as positive one, in which samples of the same class are erroneously pulled apart. This situation can lead to a decrease in the accuracy of the k-nn classifier. As shown in Table 1, the contrastive method SimCLR performs poorly when using k-nn classifier.
> > &emsp; Meanwhile, this situation can also occur in non-contrastive learning as illustrated in [2]. The strength of method constraints on embedding can influence sample distribution. our INTL introduces a moderate equal-eigenvalues constraint on the embedding. We conject that this moderate constraint on embedding results in a tendency for a well-balanced similarity among different samples within the dataset: samples from distinct categories can be effectively distinguished, and samples from the same category are not heavily pushed apart. This equilibrium in the local structure of INTL's representation provides effective guidance for sample classification, contributing to its superior performance in 5-nn accuracy.
> >
> > **some typos:** We thank the reviewer and have fixed all these in the revised manuscript.
> >
> > > **References**
> > > [1] Ghosh, Arna, et al. Investigating power laws in deep representation learning. arXiv preprint arXiv:2202.05808 (2022).
> > > [2] Bobby He and Mete Ozay. Exploring the gap between collapsed and whitened features in selfsupervised learning. In ICML, 2022.
> > > [3] Xi Weng, Lei Huang, Lei Zhao, Rao Muhammad Anwer, Salman Khan, and Fahad Khan. An investigation into whitening loss for self-supervised learning. In NeurIPS, 2022.

---

> > > ### Author Response · Authors · 2023-11-22
> > > **Official Comment by Authors**
> > >
> > > Dear Reviewer AFis,
> > >
> > > We recognize that the timing of this discussion period may not align perfectly with your schedule, yet we would greatly value the opportunity to continue our dialogue before the deadline approaches.
> > >
> > > Could you let us know if your questions have been adequately addressed? If not, please feel free to raise them, and we are more than willing to provide further clarification; if you find that your concerns have been resolved, we would appreciate if you could re-consider the review score.
> > >
> > > We hope that we have resolved all your questions, but please let us know if there is anything more.
> > >
> > > Best wishes to you!

---

### Official Review · Reviewer_YNa9 · 2023-11-02

**Soundness:** 3 good
**Presentation:** 3 good
**Contribution:** 3 good
**Rating:** 6
**Confidence:** 4

**Summary:**

This paper presents a new approach, Spectral Transformation (ST), for self-supervised learning, and proposes a new training algorithm named IterNorm with trace loss (INTL). The basic idea of the paper is to balance the spectrum of the covariance matrices for the learned features which is often ill-posed. Theoretical and empirical results are provided as well for demonstrating the performance.

**Strengths:**

Clear writing with many experimental results.

**Weaknesses:**

To me, I do not see any obvious weakness of the proposed approach. Motivated by the whitening, the paper presents a nice and logical development of the approach. However, I do not see a very strong point neither that can make this paper stand out compared with the literature. I suggested the authors to further emphasize the key contributions: What really makes your approach better than others such as MoCo and SimCLR? How about computational speed (this seems not to be discussed in both paper and appendix)?

I’d like to increase my rate if the authors can convince me at this point.

**Questions:**

see my comment

---

> ### Author Response · Authors · 2023-11-14
> **Thanks sincerely for your encouraging words and constructive comments.**
>
> ## **Responses (1/2)**
>
> **Question 1:** What really makes your approach better than others such as MoCo and SimCLR?
>
> **Response:** Following your constructive suggestions, we further emphasize the advantages of our proposed INTL compared to other methods step by step.
> 1. **Without negative pairs: Our INTL eliminates the need for negative example construction, mitigating potential challenges inherent in contrastive learning which enlarges the dissimilarity among samples belonging to the same latent label.**
> > &emsp; In contrastive methods like MoCo and SimCLR, negative samples play an important role and need to be well-designed. However, the designed negative samples are likely to have the same latent label as the positive samples. In this case, contrastive methods will enlarge the dissimilarity among these samples, and could potentially lead to the disruption of the integrity of the potential manifold during the training process [1]. As a non-contrastive approach, our INTL circumvents this issue by eliminating the need for negative examples.
>
> 2. **Robust to hyperparameters: Our INTL demonstrates robustness across various hyperparameters compared to the non-contrastive counterparts, making it adaptable to diverse application scenarios.**
> > &emsp; Although other non-contrastive methods also eliminate the need for constructing negative examples, many of these approaches display sensitivity to hyperparameters (including training parameters like batch size and algorithm parameters like penalty coefficient), posing challenges in their adaptation to diverse scenarios.
> > &emsp; Some methods are sensitive to training parameters. For instance, BYOL, SwAV, and W-MSE are sensitive to batch size; they require a large training batch size to work well [2, 3]. As dimension de-correlation methods, Barlow Twins and VICReg are sensitive to embedding dimensions; they require a large embedding dimension to work well [4, 5]. These requirements increase the demand for computational resources.
> > &emsp; Moreover, certain SSL methods are sensitive to algorithm parameters and necessitate substantial parameter tuning when applied to diverse datasets or network architectures. For example, SimSiam, DINO, and SwAV exhibit poor performance on CIFAR-100 and ImageNet-100 when using the parameters provided in their settings on ImageNet-1K [6].
> > &emsp; In contrast, our INTL are robust to hyperparameters. It achieves commendable performance with small batch size (Table 2). It also exhibits robustness to embedding dimensions (Figure 8). Notably, INTL does not necessitate parameter tuning (the iteration number $T$ of IterNorm and the coefficient $\beta$ are fixed when applied to different datasets and network architectures), as shown in Appendix C. These characteristics underscore its adaptability and effectiveness across diverse scenarios.
>
> 3. **Moderate constraints: Our INTL imposes a moderate constraint on embedding which results in a satisfactory spectrum distribution, thereby endowing the representation with good generalization capabilities.**
> > &emsp; Recent theoretical work [7] proposes a power law to analyze the relationship between eigenspectrum and representation quality. Meanwhile, [8] demonstrates that the degree of feature whitening influences generalization. Both of these works emphasize that a steep spectrum diminishes representation quality, a smooth spectrum compromises generalization, while **a moderate spectrum ensures optimal generalization ability**. SSL methods impose their constraints on embedding to prevent collapse, and the strength of these constraints directly influences the spectrum distribution.
> > &emsp; Hard whitening methods, such as W-MSE and CW-RGP, impose a full-rank constraint on embedding [2]. However, [2] indicates that the full-rank constraint is too weak to achieve optimal performance, leading to the proposal of a random group partition technique to enhance constraints and improve results. Nevertheless, this enhancement is unpredictable and comes with additional computational costs, especially on large datasets.
> > &emsp; Soft whitening methods, like Barlow Twins and VICReg, impose a strong whitening constraint on embedding, urging the covariance matrix of the embedding to be identity. The results in Table 3 show that these methods (Barlow Twins) perform even worse than hard whitening methods (W-MSE) in transfer learning, which indicates that excessive constraints have already compromised their generalization ability for downstream tasks.
> > &emsp; In contrast, our INTL provides a moderate equal-eigenvalues constraint on embedding, which is stronger than hard whitening (full-rank constraint) but weaker than soft whitening (whitening constraint). Our INTL performs much better than whitening methods in transfer learning, which indicates this moderate constraint enables the representation to attain a favorable spectrum distribution, consequently endowing it with good generalization capabilities.

---

> ### Author Response · Authors · 2023-11-14
> **We sincerely appreciate your time in reading the paper**
>
> ## **Responses (2/2)**
>
> 4. **Theoretical guarantee: Our INTL is supported by theoretical evidence to avoid collapse.**
> > &emsp; Although some non-contrastive methods such as BYOL, SimSiam, and DINO work well to avoid collapse, it remains unclear how these asymmetric networks effectively prevent collapse without the inclusion of negative pairs. Instead, our INTL is theoretically guaranteed to avoid collapse and our theoretical analysis presents a new thought in demonstrating how to avoid dimensional collapse.
>
> In summary, **our INTL method is free to negative pairs, robust to hyperparameters,  moderate to eig-spectrum constraints, and guaranteed to avoid collapse**. These characteristics of INTL make it have the potential to stand out in self-supervised learning. We hope the detailed response above can convince you. Thanks!
>
> **Question 2:** How about computational speed (this seems not to be discussed in both paper and appendix)?
>
> **Response:** We have reported the computational cost of INTL (time and GPU memory requirements of our implementation for INTL trained per epoch on ImageNet with ResNet-50) in Table 9 of the appendix. Our INTL with EMA requires a total of around 23.6 GB GPU memory and 24min46s running time per epoch, which is comparable to 20 GB memory and 23min11s running time per epoch required by MoCo-v2 which also uses EMA. Meanwhile, compared to other methods that use an additional predictor (such as BYOL), or use eigen-decomposition (such as W-MSE), our INTL requires less memory and computation time.
>
> > **References**
> > [1] HaoChen, J.Z., Wei, C., Gaidon, A., Ma, T.: Provable guarantees for self-supervised deep learning with spectral contrastive loss. In: NeurIPS (2021)
> > [2] Xi Weng, Lei Huang, Lei Zhao, Rao Muhammad Anwer, Salman Khan, and Fahad Khan. An investigation into whitening loss for self-supervised learning. In NeurIPS, 2022.
> > [3] Xinlei Chen and Kaiming He. Exploring simple siamese representation learning. In CVPR, 2021.
> > [4] Jure Zbontar, Li Jing, Ishan Misra, Yann Lecun, and Stephane Deny. Barlow twins: Self-supervised learning via redundancy reduction. In ICML, 2021.
> > [5] Adrien Bardes, Jean Ponce, and Yann LeCun. Vicreg: Variance-invariance-covariance regularization for self-supervised learning. In ICLR, 2022.
> > [6] Victor Guilherme Turrisi da Costa, Enrico Fini, Moin Nabi, Nicu Sebe, and Elisa Ricci. solo-learn: A library of self-supervised methods for visual representation learning. Journal of Machine Learning Research, 23(56):1–6, 2022.
> > [7] Ghosh, Arna, et al. Investigating power laws in deep representation learning. arXiv preprint arXiv:2202.05808 (2022).
> > [8] Bobby He and Mete Ozay. Exploring the gap between collapsed and whitened features in selfsupervised learning. In ICML, 2022

---

> > ### Author Response · Authors · 2023-11-22
> > **Official Comment by Authors**
> >
> > Dear Reviewer YNa9,
> >
> > We recognize that the timing of this discussion period may not align perfectly with your schedule, yet we would greatly value the opportunity to continue our dialogue before the deadline approaches.
> >
> > Could you let us know if your questions have been adequately addressed? If not, please feel free to raise them, and we are more than willing to provide further clarification; if you find that your concerns have been resolved, we would appreciate if you could re-consider the review score.
> >
> > We hope that we have resolved all your questions, but please let us know if there is anything more.
> >
> > Best wishes to you!

---

### Meta-Review · Area_Chair_gwKx · 2023-12-06

**Metareview:**

The paper introduces a novel approach named Spectral Transformation (ST) for self-supervised learning, with a specific focus on a new training algorithm called IterNorm with trace loss (INTL). The main objective of the paper is to address the issue of balancing the spectrum of the covariance matrices for learned features, a problem often encountered in self-supervised learning. All reviewers rated the paper as marginally above the acceptance threshold. The reviewers agree the paper presents a novel approach to modulate the spectrum of embeddings and prevent dimensional collapse, an important issue in contrastive learning. The paper has solid theoretical grounding and significant empirical improvements, especially in transfer learning tasks on real-world datasets. The reviewers also point out the paper should be further enhanced to emphasize the unique contributions of their approach compared to existing literature, to provide more detailed comparisons with whitening methods, to clarify the motivation and intuitive aspects of their approach, and to consider including missing baseline comparisons. The AC checked the materials, and agrees with the reviewers to accept the paper.

**Justification For Why Not Higher Score:**

The reviewers point out the paper should be further enhanced to emphasize the unique contributions of their approach compared to existing literature, to provide more detailed comparisons with whitening methods, to clarify the motivation and intuitive aspects of their approach, and to consider including missing baseline comparisons.

**Justification For Why Not Lower Score:**

The reviewers agree the paper presents a novel approach to modulate the spectrum of embeddings and prevent dimensional collapse, an important issue in contrastive learning. The paper has solid theoretical grounding and significant empirical improvements, especially in transfer learning tasks on real-world datasets.

---

### Decision · Program_Chairs · 2024-01-16

Accept (poster)